# Regulation of cytokine signaling through direct interaction between cytokine receptors and the ATG16L1 WD40 domain

Inmaculada Serramito-Gómez [1], Emilio Boada-Romero [1], Raquel Villamuera[1], Álvaro Fernández-Cabrera[1], José Luis Cedillo[1], Ángela Martín-Regalado[1], Simon Carding[2], Uli Mayer[3], Penny P. Powell [4], Thomas Wileman [2], Irene García-Higuera[5] & Felipe X. Pimentel-Muiños [1,6]✉

ATG16L1, an autophagy mediator that specifies the site of LC3 lipidation, includes a C-terminal domain formed by 7 WD40-type repeats (WD40 domain, WDD), the function of which is unclear. Here we show that the WDD interacts with the intracellular domain of cytokine receptors to regulate their signaling output in response to ligand stimulation. Using a refined version of a previously described WDD-binding amino acid motif, here we show that this element is present in the intracellular domain of cytokine receptors. Two of these receptors, IL-10RB and IL-2Rγ, recognize the WDD through the motif and exhibit WDD-dependent LC3 lipidation activity. IL-10 promotes IL-10RB/ATG16L1 interaction through the WDD, and IL-10 signaling is suboptimal in cells lacking the WDD owing to delayed endocytosis and inefficient early trafficking of IL10/IL-10R complexes. Our data reveal WDD-dependent roles of ATG16L1 in the regulation of cytokine receptor trafficking and signaling, and provide a WDD-binding motif that might be used to identify additional WDD activators.

[1] Instituto de Biología Molecular y Celular del Cáncer, Centro de Investigación del Cáncer, Consejo Superior de Investigaciones Científicas (CSIC)-Universidad de Salamanca, Campus Miguel de Unamuno, 37007 Salamanca, Spain. [2] Quadram Institute Bioscience, Norwich Research Park, Norwich NR4 7UQ, UK. [3] School of Biological Sciences, University of East Anglia, Norwich Research Park, Norwich NR4 7TJ, UK. [4] Norwich Medical School, University of East Anglia, Norwich Research Park, Norwich NR4 7TJ, UK. [5] Departamento de Biología Molecular, Centro de Biología Molecular Severo Ochoa (UAM-CSIC), Universidad Autónoma de Madrid, C/Nicolás Cabrera, 1, 28049 Madrid, Spain. [6] Present address: Centro de Biología Molecular Severo Ochoa (CSIC-UAM), Nicolás Cabrera, 1, 28049 Madrid, Spain. ✉email: fxp@usal.es

Macroautophagy (hereafter referred to as autophagy) is a catabolic pathway that degrades cytoplasmic components to maintain a healthy cellular homeostasis[1,2]. This process is mediated by a group of proteins called autophagy-related proteins (ATGs)[3,4]. Judging from the pathological phenotypes displayed by ATG-deficient mice, autophagy prevents a number of diseases[5,6], suggesting a possible value as a therapeutic target[7]. However, different ATGs have been involved in activities unrelated to the canonical autophagic pathway[8–10]. The contribution of these atypical functions to the prevention of the pathological phenotypes that arise in ATG-deficient conditions is currently unclear. Therefore, identification and characterization of such activities is important to establish exactly how the different ATGs prevent disease and how modulation of their activity could be used as a valid therapeutic strategy.

ATG16L1, a critical autophagic effector that specifies the site of LC3 lipidation[11], is a particularly interesting example of this issue because the canonical and unconventional functions of the molecule appear to involve different domains. Thus, ATG16L1 includes a C-terminal domain formed by 7 WD40-type repeats (WD40 domain, WDD)[12] whose physiological functions have remained unclear. This is because the N-terminal region of ATG16L1 binds the LC3 lipidation machinery (the dimer ATG5-12)[12,13] and suffices to sustain canonical autophagy[14–16], whereas the C-terminal WDD (residues 320–607) is irrelevant for this basic route. Instead, the WDD is critical for a collection of poorly characterized unconventional activities where LC3 lipidation is activated in single membrane, atypical subcellular localizations[15–17], including phagosomal vacuoles in a process called LC3-associated phagocytosis (LAP)[16]. The WDD is recognized by upstream activators including a 4-aminoacid, tyrosine-based motif originally identified in the transmembrane molecule TMEM59 ([YWF]-X$_3$-[ED]-X$_4$-[YWF]-X$_2$−L)[15,17]. Engagement of ATG16L1 through this motif allows TMEM59 to label its own single-membrane endosomes with LC3, thus increasing their fusion with the lysosomal compartment[17]. Therefore, the WDD appears to act as an interaction module for proteins that enable the LC3 lipidation capacity of the N-terminal domain in atypical subcellular localizations. We reasoned that the WDD-binding motif could help identify physiological activators of the WDD and novel unconventional functions of ATG16L1 carried out through this region. However, in its original configuration the motif is too degenerated for productive comparisons with the protein sequence bank.

Here, we describe a strategy to improve the WDD-binding signature using systematic mutagenesis and in vitro binding studies based on peptide microarrays. This approach leads to a refined motif version that facilitates the unbiased identification of candidate WDD-interacting molecules. Notably, we find the improved motif represented in the intracellular region of a number of type-I transmembrane proteins, including different cytokine receptors. We present data showing that one of the receptors, IL-10RB, relies on the WDD for proper intracellular trafficking and optimal signaling in response to IL-10.

## Results

**Refinement of the WDD-binding motif.** We used peptide microarrays to explore the impact of all possible aminoacids at all positions of the active TMEM59 peptide (residues 268–280; Fig. 1a) on the interaction with the WDD. Previous studies indicate that this approach is feasible, since the 268–280 peptide immobilized onto a glass slide is able to specifically bind WDD-containing constructs[15]. Arrays including the unmodified element, as well as a collection of mutant peptides where every residue was changed to all possible aminoacids were probed with a recombinant protein that includes the WDD (GST-HA-

ATG16L1$_{231–607}$). Average binding data obtained for each mutant were normalized against the binding value provided by the wild-type peptide, and an arbitrary threshold was established to categorize as permissive or prohibited those substitutions with an incomplete impact on the binding event (Fig. 1b; Supplementary Fig. 1a, b). Data obtained for high and low ligand concentrations were processed separately because some mutations qualified as permissive or prohibited depending on the dose of the recombinant protein (see, for example, S271N; Fig. 1b). The resulting heat-maps confirmed Y268, Y277, and L280 as the most restrictive positions, and revealed a new island of restriction around aminoacid K273 (Fig. 1b). To establish how the distance between critical residues may influence the interaction, we resorted to arrays including peptides where such distances were altered by introducing extra residues (alanine) or by deleting native aminoacids. However, changing the spacing between important positions implicitly modifies the aminoacid composition of the relevant stretch, and therefore there is an uncertainty as to whether any impact in binding would originate from one or the other varying parameters (distance vs. composition). We reasoned that, if a spacing-modified peptide fully retains the ability to bind the ligand, the relevant distance should be considered as permissive irrespective of whether or not less stringent modifications may inhibit the interaction. The results indicate that the motif can accommodate substantial distance flexibility between residues Y268-E272 and E272-Y277, whereas the distance between Y277 and L280 cannot be modified without impairing the interaction (Fig. 1c). Previous data showed that a negatively charged residue (D or E) in position 272 is required for the LC3-lipidation activity of TMEM59 but not for LC3-I recruitment[17], so to facilitate identification of a motif able to promote LC3 lipidation, the presence of D or E was imposed at this position. Collectively, these studies led to a refined WDD-binding motif version where most positions are defined by a discrete set of allowed aminoacids (Fig. 1d).

Bioinformatic comparisons using the Prosite algorithm showed that a wide variety of type-I transmembrane proteins include this element in their intracellular region (Supplementary Data 1 and 2). Transmembrane proteins are particularly interesting in this context, because their engagement of the WDD would facilitate direct access of the ATG16L1-5-12 complex to a membrane source for LC3 lipidation. Interestingly, the collection included 27 cytokine receptors (Supplementary Data 3 and 4), suggesting that, similar to TMEM59, their intracellular trafficking and/or functional features might be modulated by ATG16L1 through the WDD[15,17].

**IL-2Rγ and IL-10RB include a functional WDD-binding motif.** To explore this idea, we implemented a sequential selection approach directed to identify candidate receptors exhibiting WDD-dependent mechanistic traits. We first tested a total of 12 receptors for binding to the WDD in co-precipitation studies. 11 of them were able to interact with the WDD (Fig. 2a; Supplementary Fig. 2a), but only 5 did so specifically, without also binding the N-terminal region of ATG16L1 (Fig. 2a). We focused on these to evaluate their ability to induce LC3 lipidation. Previous studies indicate that overexpression of TMEM59 induces unconventional, WDD-dependent LC3 lipidation[15,17], so we used this simple property to quickly identify those receptors whose interaction with the WDD may result in activation of the LC3 lipidation complex. Transfection of IL-2Rγ, IL-10RB, and IL-1RL1 (IL-33R) into HEK-293T cells caused robust lipidation of co-transfected AU-LC3 (Supplementary Fig. 2b), whereas IL-18R1 and IL-18RAP had a marginal activity (Supplementary Fig. 2b). To examine if the autophagic activity of IL-2Rγ,

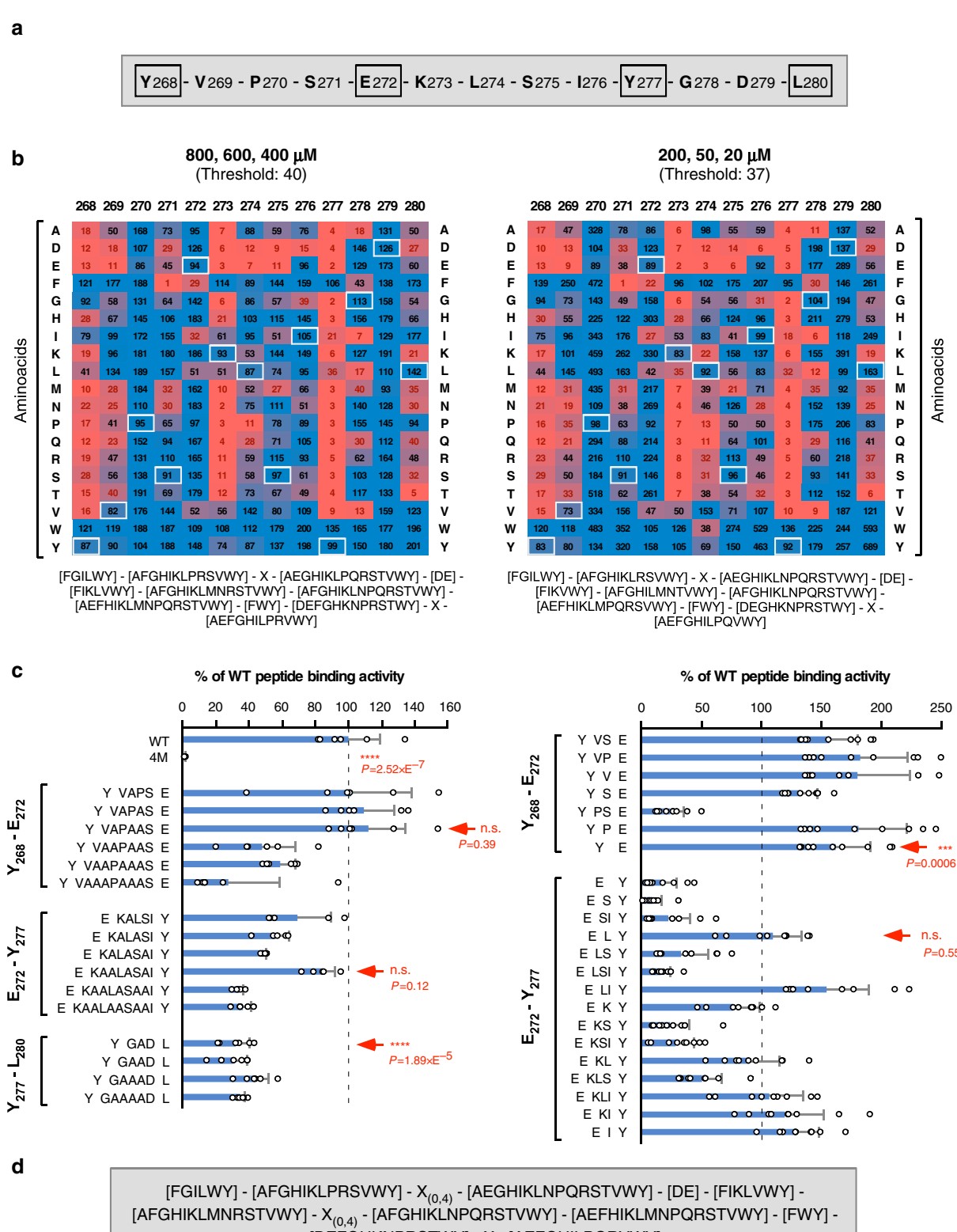

IL-10RB, and IL-33R requires the WDD, we resorted to HCT116 cells expressing separated Nt (1–299) and Ct (300–607) domains of ATG16L1. In these cells the effector Nt domain is physically disconnected from the WDD interaction platform, thus impairing LC3 lipidation in response to WDD engagement[15]. Notably, while expression of IL-2Rγ, IL-10RB, and IL-33R triggered LC3 lipidation in cells harboring full-length ATG16L1, this activity was blocked in cells expressing separated Nt and Ct domains

(Fig. 2b). This result points to a critical role of the WDD in the LC3 lipidation activity triggered by these receptors. In addition, we found significant colocalization between IL-2Rγ or IL-10RB and co-transfected GFP-LC3, either in the absence (IL-2Rγ) or presence (IL-10RB) of bafilomycin A1 to minimize receptor degradation (Fig. 2c). This result indicates that at least part of the lipidated LC3 that is generated by both receptors ends up decorating their own intracellular vesicles. This is expected from an

**Fig. 1 WDD-binding motif refinement. a** WDD-binding peptide present in TMEM59 intracellular domain (268–280). Critical motif positions are marked by squares. **b** Peptide microarray studies to identify permissive aminoacids for WDD binding. Arrays containing the indicated peptide mutants were developed with GST-HA-ATG16L1$_{231-607}$ to obtain crude binding values. Heatmaps display binding data expressed as the average percentage of the wild-type binding activity that is retained by each mutant. Left and right heatmaps show average data from assays performed at high (800, 600, 400 nM) and low (200, 50, 20 nM) ligand concentrations, respectively (3 experimental points, $n = 8$; statistics in Supplementary Fig. 1a). The color scale stretches continuously from low (red) to high (blue) binding activity. The indicated threshold values were arbitrarily established as the average binding activity displayed by variants that inhibit the interaction beyond 5%. Black/red figures indicate values above/below the threshold, thus establishing permissive/prohibited substitutions, respectively. White squares indicate native aminoacids in the wild-type peptide. Permissive residues for all positions are displayed below the heatmaps. **c** Microarray studies to establish the distance flexibility between critical residues. Microarrays including peptides with the indicated increased (left) or decreased (right) distances between Y268-E272, E272-Y277, and Y277-L280 were developed as in **a**. 4 M indicates a control peptide where critical residues are mutated to alanine (Y268A, E272A, Y277A, L280A). Shown are average binding activities processed as in **a** (−/+ s.d.). Data originate from 2 (left; 800, 400 nM ($n = 6$) except for E272-Y277 where $n = 5$) or 3 (right; 800, 600, 400 nM; $n = 9$) experimental points (n.s: $P > 0,05$; ***$P < 0.001$; ****$P < 0.0001$, two-sided Student's $t$-test). Arrows indicate data defining the maximal (left) and minimal (right) distances allowed for each stretch. Thus, peptides YVAPAASE (Y268-E272) and EKAALASAIY (E272-277) (left), or YE (Y268-E272) and ELY (E272-277) (right), fully retain WDD-binding activity, indicating that 6 and 8 residues (left), or 0 and 1 (right), respectively, are permitted. Minimal variations in Y277-L280 spacing (YGADL; left) inhibit binding, so the native 2-residue distance was considered critical and deletions were not tested. **d** Inclusive WDD-binding motif integrating permissive residues and distance flexibility.

unconventional autophagic response induced by direct engagement of ATG16L1 and the LC3-lipidation complex in the vicinity of a membrane source for LC3 lipidation, as previously described[15,17]. On the contrary, IL-33R showed poor colocalization with GFP-LC3 that was not increased by bafilomycin A1 (Supplementary Fig. 2c), so this receptor was not further analyzed. The interaction of IL-2Rγ and IL-10RB with ATG16L1 was solidified by additional co-immunoprecipitation studies performed in different conditions (Supplementary Fig. 3a, b).

Importantly, mutational inactivation of the WDD-interaction motifs present in IL-2Rγ and IL-10RB (Fig. 3a) impaired binding of both receptors to the WDD (Fig. 3b) and full-length ATG16L1 (Supplementary Fig. 4a, b), as well as their ability to induce LC3 lipidation (Fig. 3c) and colocalization with GFP-ATG16L1 in co-transfection experiments (Fig. 3d). These data indicate that both IL-2Rγ and IL-10RB include functional motifs able to mediate their interaction with ATG16L1 to induce LC3 lipidation. More generally, these results show that the refined motif is able to identify real interactors of the WDD. Although these rudimentary overexpression assays were helpful to select candidate receptors that exhibit WDD-dependent functional features, studying how the biological activity of the selected molecules may be influenced by the WDD requires the use of more sophisticated experimental systems where candidate activity can be assessed in the presence of the relevant co-receptors and in response to their natural ligands.

**Optimal IL-10 receptor signaling requires the WDD**. To explore how the interaction with ATG16L1 and the WDD may shape the physiological properties of motif-containing cytokine receptors, we focused on IL-10RB, the receptor of a prominent anti-inflammatory cytokine (IL-10)[18]. Both ATG16L1[19] and the WDD[20] were previously shown to counter inflammation in different experimental paradigms, so we hypothesized that the interaction between IL-10RB and ATG16L1 might be important to sustain the anti-inflammatory function of IL-10. To develop a suitable experimental system where we could unambiguously assess the role of the WDD in IL-10R physiology, we restored $Atg16l1^{-/-}$ mouse embryonic fibroblasts (MEFs) with different versions of ATG16L1 and reconstituted the whole IL-10 signaling system in these cells. $Atg16l1^{-/-}$ MEFs were first retrovirally transduced with a STAT3-luciferase reporter system and a vector expressing STAT3, the main transcription factor induced by IL-10[18]. Cells were then restored with either full-length ATG16L1 (FL) or just the N-terminal domain (1–299; Nt), and subsequently

transduced with vectors expressing IL-10Rs A and B. Although the Nt domain was expressed at slightly lower levels compared to FL ATG16L1 (Supplementary Fig. 5a), the canonical autophagic flux was recovered to a similar extent in both cell lines (Supplementary Fig. 5b). Expression of both IL-10Rs was comparable between cells harboring FL or Nt constructs (Supplementary Fig. 5c). However, cells lacking ATG16L1 showed increased levels of IL-10RA and B (Supplementary Fig. 5c), an unexpected effect whose underlying mechanism is currently unknown. Surface exposure of both IL-10Rs was also similar in FL and Nt cells (Supplementary Fig. 5d, e), thus allowing direct comparison of IL-10 activity between both cell lines to specifically evaluate the role of the WDD. The engineered cells became responsive to IL-10 (Supplementary Fig. 5f).

Co-immunoprecipitation studies revealed that IL-10 induces the interaction between IL-10RB and FL-ATG16L1 (Supplementary Fig. 6a), an association that was strengthened by the concomitant presence of lysosome inhibitors to prevent receptor degradation (Supplementary Fig. 6b). No co-precipitation was detected in cells expressing the Nt construct (Fig. 4a), indicating that the WDD mediates this interaction. Surprisingly, immunofluorescence assays showed a very low level of colocalization between endocytosed IL-10RB and LC3 in IL-10-treated cells (Supplementary Fig. 7a). However, pre-treatment with the lysosomal inhibitor bafilomycin A1 revealed substantial IL-10RB/LC3 colocalization in cells expressing FL ATG16L1 (Supplementary Fig 7b), likely because impairment of lysosomal degradation slows down IL-10RB trafficking and facilitates detection of transient LC3 decoration events that may be difficult to observe. Interestingly, IL-10RB/LC3 colocalization was poorer in cells expressing the Nt domain (Supplementary Fig. 7b), arguing that IL-10R endosomes reach transient LC3-positive compartments more efficiently through a mechanism involving the WDD.

STAT3-dependent luciferase activity assays in response to IL-10 showed that IL-10 signaling is defective in cells expressing the Nt domain compared to those including FL ATG16L1 (Fig. 4b), whereas the same activity induced by Leukemia Inhibitory Factor (LIF) through endogenous receptors remains unaffected (Supplementary Fig. 8a). The impaired IL-10 signaling output exhibited by Nt cells was accompanied by reduced early cytoplasmic phosphorylation of STAT3 (cytoplasmic extracts) and diminished nuclear translocation of P-STAT3 (nuclear extracts), both in response to continuous IL-10 treatment (Supplementary Fig. 8b) and, more prominently, in pulse-chase experiments (Fig. 4c). The latter approach also revealed reduced levels of IL-10 in

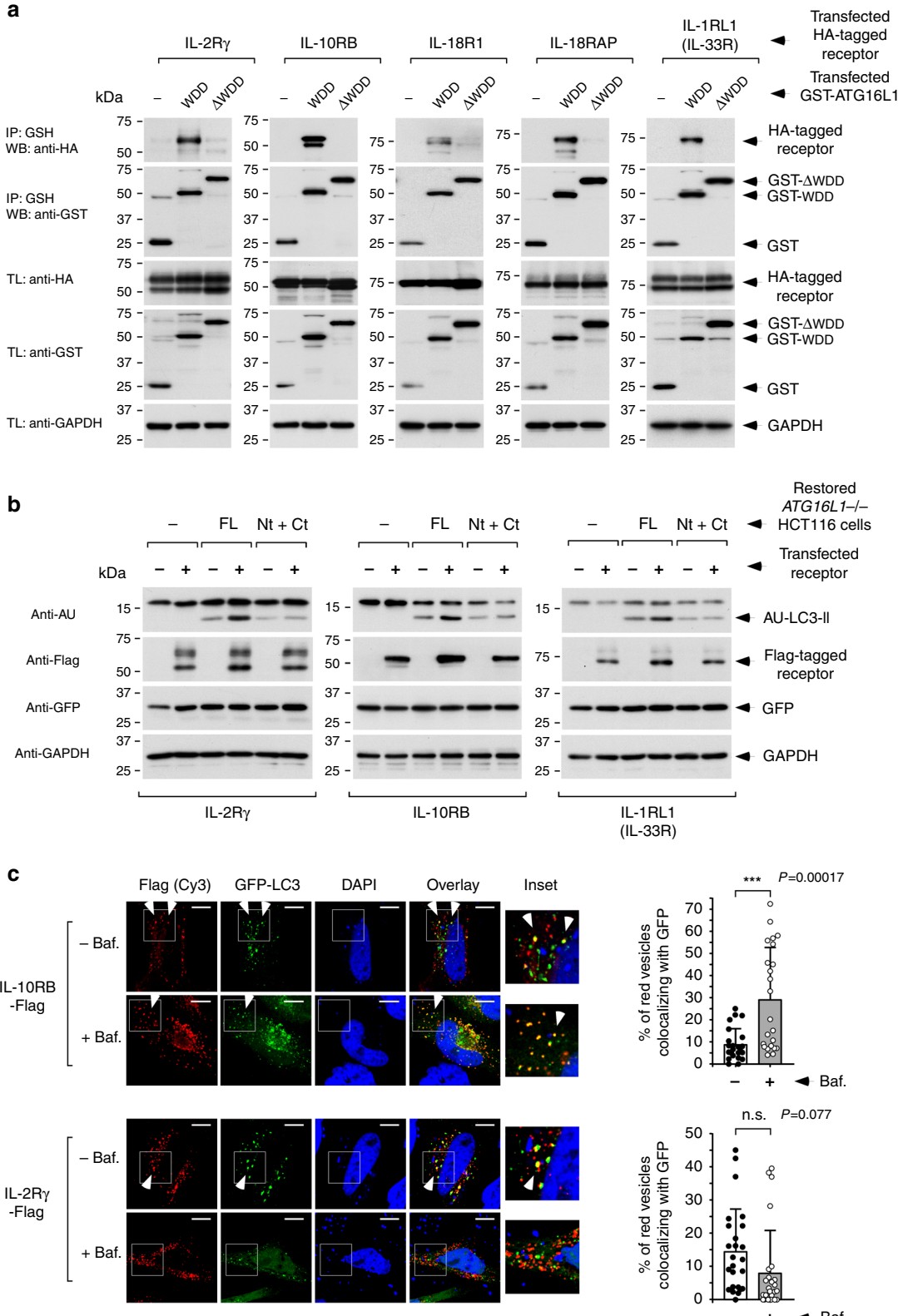

cytoplasmic extracts of cells expressing the Nt construct (Fig. 4c), pointing to a defect in endocytosis and/or trafficking of IL-10/IL-10R complexes in the absence of the WDD.

**Defective IL-10RB early trafficking in the absence of WDD.** To analyze the intracellular trafficking of IL-10Rs in response to

IL-10, we carried out immunofluorescence studies using an anti-Flag antibody that recognizes Flag-tagged IL-10RB. Compared to cells expressing FL ATG16L1, those harboring the Nt fragment presented substantially reduced numbers of IL-10RB-Flag-positive cytoplasmic vesicles early after IL-10 exposure (5 and 15 min time points; Fig. 4d). Similar results were obtained using fluorescent dextran as an endocytic tracker (Supplementary

**Fig. 2 Selection of candidate receptors showing WDD-dependent functional features. a** Co-immunoprecipitation studies to identify receptors able to specifically interact with the WDD. HEK-293T cells were co-transfected with tagged forms of the indicated cytokine receptors along with constructs including the WDD (320–607) or ΔWDD (1–319) fused to GST, as shown. Cells were lysed 36 h later and subjected to GST immunoprecipitation with agarose beads coupled to GSH. Shown are Western-blots against the indicated molecules. IP: immunoprecipitate; TL: total lysate. **b** Evaluation of WDD-dependent LC3 lipidation activity. ATG16L1−/− HCT116 cells restored with the indicated versions of ATG16L1 (full-length (FL) or separated N-terminal (Nt; 1–299) and C-terminal (Ct; 300–607) domains) were transfected with the shown tagged cytokine receptors along with plasmids expressing AU-LC3 and GFP. Cells were lysed 36 h post-transfection for Western-blotting against the indicated molecules. The expression levels of co-transfected GFP remain unchanged, indicating equal transfection efficiency. **c** Colocalization between transfected cytokine receptors and co-expressed GFP-LC3. HeLa cells were transfected with the indicated tagged cytokine receptors and a plasmid expressing GFP-LC3. Cells were treated with bafilomycin A1 (75 nM, as indicated) during the last 6 h of culture, fixed 36 h post-transfection, stained with an anti-Flag antibody and scored for colocalization between the Flag signal (red) and GFP-LC3. Shown are representative confocal pictures (left) where the white arrows indicate colocalization events. Graphs (right) display the percentage of Flag-positive vesicles colocalizing with the GFP-positive signal in the different conditions −/+ s.d. (n = 23 cells for IL-10RB; n = 25 cells for IL-2Rγ; n.s: P > 0,05; ***P < 0.001, two-sided Student's t-test).

Fig. 9a). The dissimilar numbers of Flag-positive vesicles in FL and Nt cells were not normalized by blocking lysosomal function with bafilomycin A1 (Supplementary Fig. 9b), suggesting that they are not caused by a different degradation rate in the lysosomal compartment, but likely by reduced endosome formation in Nt cells. These results point to the notion that absence of the WDD impairs IL-10R endocytosis and/or endosome generation in Nt cells.

To explore the endocytic route followed by IL-10/IL-10R endosomes in both cell lines, we carried out immunofluorescence colocalization studies between IL-10RB-Flag and early (EEA1) and late (LAMP1 and Rab7) endocytic markers. Interestingly, the degree of colocalization between endocytosed IL-10RB and EEA1 was reduced in cells expressing the Nt construct, not only 5 min post-IL-10 exposure (where Nt cells show almost no IL-10RB endosomes and therefore this parameter is probably irrelevant), but also when comparing time points with similar numbers of IL-10RB-positive endosomes (FL vs. Nt at 15 min of IL-10 treatment, or FL at 5 min IL-10 vs. Nt at 15 min IL-10; Fig. 5a, b). Comparable results were obtained in pulse-chase IL-10 experiments (Supplementary Fig. 10a, b). However, the rate of colocalization with the late endosomal/lysosomal markers LAMP1 (continuous IL-10 treatment, Supplementary Fig. 11a, b; pulse-chase, Supplementary Fig. 12a, b) and Rab7 (continuous treatment, Supplementary Fig. 13a, b) remained comparable between FL and Nt cells in those experimental points showing a similar number of IL-10R-positive endosomes (FL vs. Nt at 15 min of IL-10 treatment, or FL 5 min vs. Nt 15 min). These results are consistent with the notion that IL-10/IL10R complexes traffic defectively to early endosomes in cells lacking the WDD, while their eventual transit to late endosomes and/or lysosomes is not substantially affected. Such comparable access to late compartments suggests again that the degradation rate of IL-10/IL-10R endosomes is unaltered by absence of the WDD, a result that is consistent with the bafilomycin data mentioned above (see Supplementary Fig. 9b). Together, these data point to the idea that the WDD allows ATG16L1 to assist IL-10RB endocytosis and trafficking to early endosomes upon activation by IL-10. Defects in this mechanism caused by absence of the WDD result in a reduced total number of endosomal vesicles carrying engaged IL-10Rs, likely explaining the blunted IL-10 signaling output exhibited by Nt cells.

An unconventional process where the autophagic machinery favors receptor recycling from endosomal compartments to the plasma membrane has recently been described (LC3-associated endocytosis, LANDO)[21]. This phenomenon relies on the autophagic protein Rubicon and, if mediated by the WDD, could increase the steady state levels of IL-10Rs at the cell surface, thus explaining the contribution of the WDD to IL-10R signaling.

However, depletion of endogenous Rubicon did not impair IL-10 signaling in MEFs expressing FL ATG16L1 (Supplementary Fig. 14a), indicating that LANDO is unlikely to be the primary mechanism through which the WDD regulates IL-10R physiology. MEFs expressing FL ATG16L1, IL-10RA and a mutated version of IL-10RB that is unable to bind the WDD (5 M) showed impaired STAT3-dependent luciferase activity induced by IL-10, and the magnitude of such defect was comparable to that observed in cells expressing the Nt domain (Supplementary Fig. 14b). This result argues that the defective signaling output caused by absence of the WDD is likely due to impairment of the IL-10RB/ATG16L1 interaction, not to indirect effects.

**Defective IL-10 signaling in macrophages lacking the WDD.** To recapitulate these results in a cellular system naturally responsive to IL-10, we resorted to the human monocytic cell line THP1, which acquires macrophage traits upon PMA treatment[22,23]. Endogenous expression of ATG16L1 was reduced in these cells using the CRISPR/Cas9 system directed to a target sequence that encodes for aminoacids included in the WDD (342–348). Expression of the Nt domain of ATG16L1 (aminoacids 1–299) in this cellular strain yielded expression levels comparable to those of the endogenous molecule (Supplementary Fig. 15a), thus allowing to test the function of the WDD in a system where the relative expression of ATG16L1 (endogenous) and the Nt domain (ectopic) is not as dissimilar as in MEFs. Both cell lines exhibited equivalent levels of basal autophagy (Supplementary Fig. 15b). Again, we observed enhanced IL-10R expression in cells lacking ATG16L1 (Supplementary Fig. 15c), but their protein levels (Supplementary Fig. 15c) and surface exposure (Supplementary Fig. 15d) were comparable in cells expressing FL or Nt ATG16L1. Treatment with IL-10 showed reduced ability in the latter strain to cause translocation of P-STAT3 to the nucleus (Fig. 6a), and to suppress the induction of inflammatory mediators in response to LPS treatment (Fig. 6b). Therefore, IL-10 signaling is also disrupted by the absence of the WDD when it is induced by endogenous IL-10Rs. In addition, such defect is also observed when the expression levels of FL-ATG16L1 and Nt are comparable, thus solidifying the results obtained in reconstituted MEFs.

Mice expressing a truncated form of ATG16L1 that lacks the WDD (E230) overcome the perinatal lethality exhibited by whole-organism ATG-deficient animals[24], and therefore can be used to study the functions of the WDD in adult tissues. Bone marrow-derived macrophages (BMDMs) from these mice differentiated normally as established by the expression levels of CD11b and F4/80 surface markers (Supplementary Fig. 16a), and expressed the expected 25 kDa fragment of ATG16L1 corresponding to aminoacids 1–230 (Supplementary Fig. 16b). Consistent with

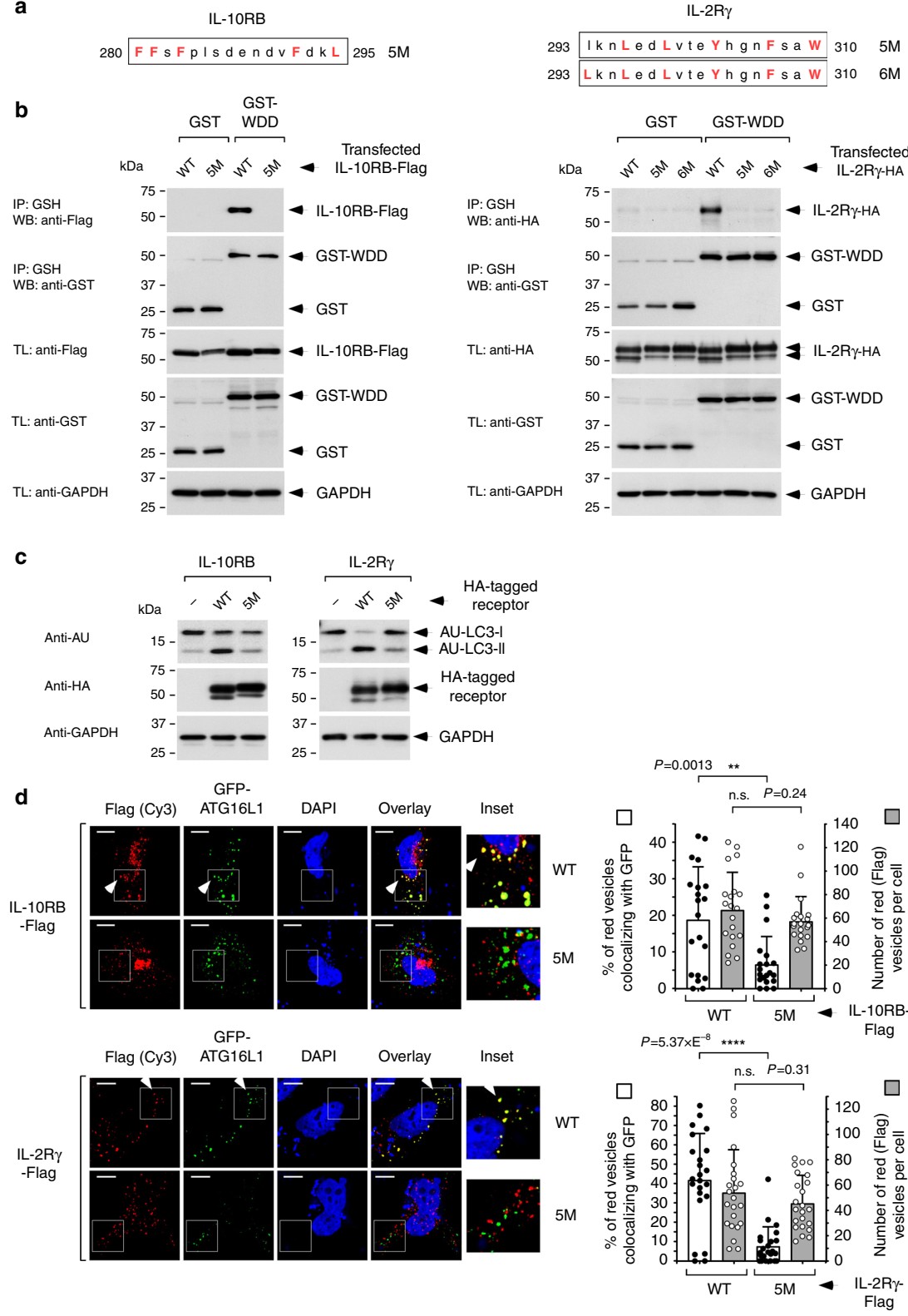

the results described above, we found reduced ability of IL-10 to promote nuclear translocation of P-STAT3 in E230 macrophages (Fig. 6c; Supplementary Fig. 16c). This defect correlated with impaired induction of Bcl3 (Fig. 6d), a direct target of IL-10 that mediates the anti-inflammatory response[25], and also with diminished suppression of TNF mRNA induction by LPS (Fig. 6e).

## Discussion

Our results show that the WDD is required for optimal IL-10 signaling by allowing ATG16L1 to interact with IL-10RB, thus favouring endocytosis and early trafficking of IL-10/IL-10R complexes. The WDD does not appear to influence the nature of the endocytic route followed by the activated IL-10Rs, nor it seems to control their degradation rate in the lysosomal

**Fig. 3 Motif-dependent activities of IL-10RB and IL-2Rγ. a** Motif-containing peptides in IL-10RB and IL-2Rγ. Critical residues highlighted in red font were mutated to alanine in order to explore motif-dependent functions. IL-2Rγ Y303 is not strictly part of the motif, but it was also mutated since Y favors binding to the WDD at all positions of the peptide. **b** Co-precipitation studies between wild-type or motif-mutated versions of the receptors and GST-WDD. HEK-293T cells were co-transfected with Flag-tagged wild-type (WT) or motif-mutated (5 M or 6 M) forms of the indicated tagged cytokine receptors and a construct expressing GST-WDD (320–607). Cells were lysed 36 h later and subjected to GST immunoprecipitation with agarose beads coupled to GSH. Shown are Western-blots against the indicated molecules. IP: immunoprecipitate; TL: total lysate. **c** Evaluation of motif-dependent LC3 lipidation activity. HEK-293T cells were transfected with the indicated versions of HA-tagged IL-10RB and IL-2Rγ (WT or 5 M) along with a plasmid expressing AU-LC3. Cells were lysed 36 h post-transfection for Western-blotting against the indicated molecules. **d** Evaluation of motif-dependent co-localization between the cytokine receptors and GFP-ATG16L1. HeLa cells were transfected with the indicated versions of Flag-tagged IL-10RB and IL-2Rγ (WT or 5 M) and a plasmid expressing GFP-ATG16L1. Cells were fixed 36 h post-transfection, stained with an anti-Flag antibody and scored for colocalization between the Flag (red) signal and GFP-ATG16L1. Shown are representative confocal pictures (left, where the white arrows indicate colocalization events), and graphs (right) representing the percentage of Flag-positive vesicles colocalizing with the GFP-positive signal −/+ s.d. (left axis, white bars, solid dots) together with the total number of Flag-positive vesicles per cell −/+ s.d. (right axis, gray bars, white dots); $n = 20$ cells for IL-10RB and $n = 23$ cells for IL-2Rγ; n.s: $P > 0,05$; $**P < 0.01$; $****P < 0.0001$, two-sided Student's $t$-test.

compartment. Instead, by favouring IL-10R endosome formation without influencing their degradation, it increases the total number of endosomes carrying activated IL-10Rs that navigate through the cytoplasm upon IL-10 stimulation. Since endosomes are recognized as an important signaling platform[26], we believe that such increased number of signaling-competent endosomes likely explains the enhanced IL-10R output that we observe in cells expressing FL ATG16L1 compared to those harboring the Nt region. However, we cannot completely exclude that the WDD may also facilitate access of IL-10R endosomes to an unknown sub-compartment that enables optimal signaling.

Exactly how the WDD and ATG16L1 promote endocytosis of IL-10Rs is currently unclear. A previously described interaction between the N-terminal region of ATG16L1 and the clathrin-heavy chain[27] might help increase the rate of endocytosis upon recruitment of ATG16L1 to the activated IL-10Rs, although a possible role of LC3 lipidation in this scenario would not be obvious. In this context ATG16L1 might function mainly as a scaffold protein, independently of its LC3 lipidation capabilities. How ATG16L1 promotes efficient trafficking of IL-10R to early endosomes is also unclear at this stage. Although LC3 has historically been linked to a degradative outcome in a variety of pathways, ranging from canonical autophagy[4] to LAP[28], it is also known to promote membrane fusion events[29], for example those involved in the secretion of lysosomal contents through the osteoclast ruffled border[30]. A role of LC3 in membrane fusion could facilitate the transition of newly formed, LC3-labeled IL-10R endosomes to EEA1-positive early compartments, a scenario that would point to a non-degradative function of LC3 in IL-10R physiology. Such unconventional function of LC3 may be implicit in this case, since its canonical degradative role would be expected to increase fusion of LC3-labeled IL-10R endosomes with the lysosomal compartment to cause earlier signal cessation, instead of an enhanced output. More generally, our data reinforce the notion that timely endocytosis and efficient early intracellular trafficking of activated IL-Rs are required factors for optimal signaling.

We believe that a similar mechanism is likely to regulate the function of other cytokine receptors that may include a functional WDD-binding motif in their intracellular region. Interestingly, we found that some of the IL-Rs that interact with the WDD also appear to bind ATG16L1 through the Nt domain, thus adding an additional layer of complexity to the IL-R/ATG16L1 functional crosstalk. The nature of such interaction with the Nt domain is currently unclear, although, as mentioned above, it could be facilitated by the clathrin-heavy chain[27]. Using the refined WDD-binding motif version we have identified a wider collection of functionally unrelated type-I transmembrane proteins that include this element in their intracellular domain, including a

variety of adhesion proteins, innate immune receptors and other uncharacterized molecules. Since their binding to the WDD is likely to depend on issues like molecular context and conformational accessibility of their different motifs, we do not expect that all of them will interact with the WDD and ATG16L1 in a natural setting. Establishing how many of them are actually able to interact with the WDD and the functional consequences of such interaction will require additional studies.

Different roles of autophagy in the control of the endocytic route have previously been described. LANDO involves labeling of internalized endosomes with LC3 to promote recycling of endocytosed receptors back to the plasma membrane instead of their lysosomal degradation, thus making them available again for ligand stimulation[21]. However, we show that depletion of Rubicon, an essential mediator of LANDO, does not impair IL-10 signaling, indicating that the WDD probably confers optimal IL-10R performance through a different mechanism. In addition, defective autophagy has been shown to induce endosome damage, EGFR accumulation in EEA1-postive vesicles and compromised signaling[31]. Absence of the WDD is unlikely to provoke a similar phenomenon, since we found reduced levels of IL-10RB colocalizing with EEA1, rather than IL-10RB accumulation in this compartment. These considerations argue that we have discovered a new molecular mechanism mediated by the WDD to regulate cytokine receptor trafficking and signaling.

A coding polymorphic allele of *ATG16L1* (rs2241880; T300A) increases the risk of Crohn disease[32−35], and impairs binding between the WDD and certain proteins including the WDD-binding motif[15], so the mechanism that we describe here might be dysregulated by ATG16L1-T300A to facilitate disease onset. Although it is difficult to anticipate how immune responses might be globally influenced by a mechanism that could regulate the function of several cytokine receptors at once, the role of the WDD in IL-10 signaling might explain some of the pro-inflammatory phenotypes exhibited by ATG16L1-deficient experimental systems[19,36]. We provide a collection of candidate receptors whose detailed and comprehensive analysis could reveal a wider role of the WDD in immune regulation and/or inter-cellular communication under a variety of physiological and pathological conditions. More generally, our improved WDD-binding motif may help identify additional molecules and mechanisms regulated by the unconventional engagement of ATG16L1 through the WDD.

## Methods

**Peptide microarray studies.** Peptide microarrays were ordered from JPT (Germany) and included each peptide printed by triplicate in three different sub-arrays (9 spots per peptide in each microarray). Peptides were immobilized through the N-terminal end using a long spacer (Tds). To purify the recombinant ligand, we transduced HEK-293T cells with a retroviral construct expressing

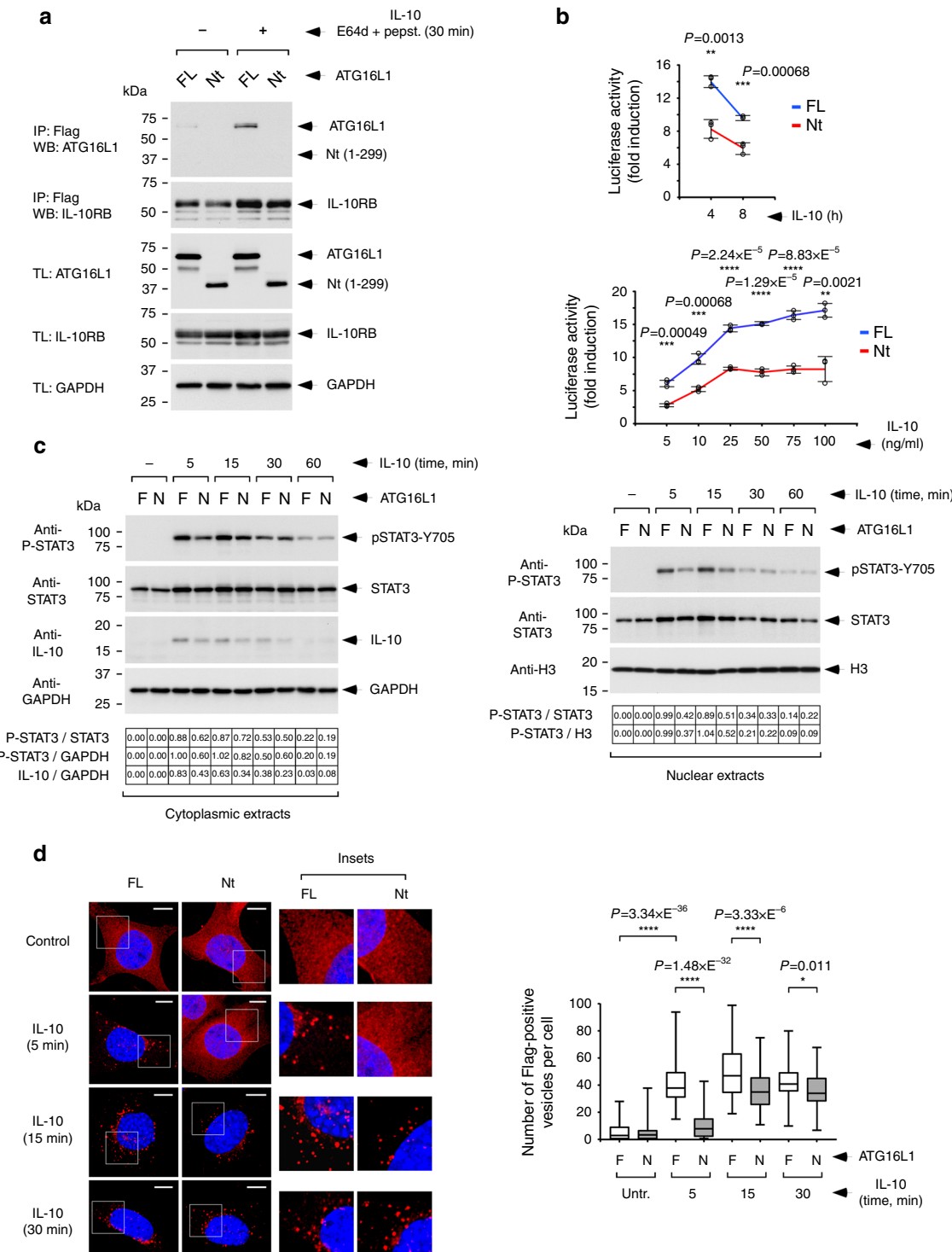

GST-HA-ATG16L1-230-607 followed by an IRES-puromycin element for selection (1 μg/ml). Cells were lysed in 1% Igepal CA-630 lysis buffer. The resulting clarified lysates were diluted to a final detergent concentration of 0,2% and incubated with GSH-Agarose beads for 4 h/4 °C. The recombinant protein was eluted from the matrix using 50 mM reduced GSH, and concentrated by centrifugation using a YM-30 Centricon filtering device (Amicon). The approximate final yield was 1–2 μg per 10-cm plate (around $10^7$ cells). Samples were run in protein gels and stained with Coomasie Blue to evaluate integrity and approximate protein quantity. A more accurate protein quantification was obtained using the DC Protein Assay kit (BioRad). Microarray slides were first blocked in a solution containing PBS/0,05% Tween-20 (PBS-T) supplemented with 5% BSA for 2 h at RT, and washed in PBS-T with gentle shaking. Ligand incubation was done in blocking solution containing the desired concentration of the recombinant protein for 2 h at RT plus overnight at

4 °C. After additional washes in PBS-T, the slides were incubated with a mixture of anti-HA (mouse monoclonal; Covance (16B12, mouse, Babco BioLegend 901501)) and anti-GST (rabbit polyclonal; Cell Signaling (rabbit, Cell Signaling 2622)) antibodies (10μg/ml each in blocking buffer) for 2 h at RT. Arrays were then developed with a solution containing a mixture of Cy5-labeled goat anti-mouse and rabbit (Jackson Immunoresearch) antibodies (1:1000 dilution in blocking buffer, 1 h at RT) followed by repeated washing in PBS-T. A final wash with MilliQ distilled water allowed removal of salt residues. Slides were scanned using a GenePix 4000B scanner (Molecular Devices) and the resulting images analyzed using the GenePix Pro 6.0 software. Data were analyzed using Microsoft Excel 1902.

**Prosite comparisons**. The WDD-binding motif resulting from peptide microarray binding studies was used to scan the landscape of human type-I transmembrane

**Fig. 4 Function of the WDD in IL-10R signaling examined in engineered MEFs.** Cells in all panels are $Atg16L1^{-/-}$ MEFs engineered to harbor a STAT3-luciferase reporter system and to express STAT3, subsequently restored with full-length ATG16L1 (FL) or a deleted version lacking the WDD (Nt, 1–299), and retrovirally transduced to express IL-10Rs A and B. **a** Co-precipitation between IL-10RB and ATG16L1 as a function of IL-10 treatment. The indicated cells (FL or Nt) were treated with IL-10 and E64d/pepstatin (10 μg/ml each) for 30 min or left untreated. Cells were then lysed and processed for anti-Flag immunoprecipitation. Shown are Western-blots against the indicated molecules. IP: immunoprecipitate; TL: total lysate. **b** Time-course and dose-response activation of STAT3-luciferase activity by IL-10. The indicated cells (FL or Nt) were treated with IL-10 for the shown times (100 ng/ml; top panel) or with the indicated doses of IL-10 (4 h; bottom panel), and lysed to measure luciferase activity. Data are expressed as the mean of fold induction of luciferase activity −/+ s.d of triplicate experimental points ($n = 3$; **$P < 0.01$; ***$P < 0.001$; ****$P < 0.0001$, two-sided Student's $t$-test) from a representative experiment. **c** Pulse-chase assay of IL-10-induced STAT3 phosphorylation. Cells (F: FL, N: Nt) were pre-incubated with IL-10 (50 ng/ml; 30 min on ice), incubated at 37 °C for the shown times and lysed to obtain cytoplasmic and nuclear extracts for Western-blotting against the indicated molecules (H3, Histone-3). Corrected densitometric quantifications are shown at the bottom of the Western-blot images. **d** Immunofluorescence assay to evaluate the number of Flag-positive intracellular vesicles in IL-10 treated cells. The indicated cells were incubated with IL-10 (50 ng/ml) for the shown times and processed for anti-Flag immunofluorescence. Shown are representative confocal pictures (left). The graph (right) displays the number of Flag-positive vesicles per cell in the different conditions. Data are presented as box-plots where the central line represents the median value, the box shows percentiles 25–75 and the whiskers include the most extreme values ($n = 56$, $n = 68$ cells for untreated and treated samples, respectively; *$P < 0.05$; ****$P < 0.0001$, two-sided Student's $t$-test).

proteins using the Option 2 (Submit MOTIFS to scan them against a PROTEIN sequence database) of the ScanProsite Tool of the ExPASy Bioinformatics Resource Portal (https://prosite.expasy.org/scanprosite/). We first generated a Fasta file including all type-I transmembrane proteins present in the human genome as annotated in the Uniprot sequence database. The database was downloaded from the Uniprot website (https://www.uniprot.org) after using the search term: locations:(location:"Single-pass type I membrane protein [SL-9905]") AND reviewed: yes AND organism:"Homo sapiens (Human) [9606]". The resulting list included 1228 proteins (in year 2017) and is provided as Supplementary Data 5. We then instructed the ScanProsite server to find in this file proteins containing the refined motif in their primary sequence. To account for motif versions including spacing distances between critical residues that are shorter than those natively present in TMEM59, we performed repeated comparisons with motif versions progressively lacking all possible combinations of the aminoacids previously identified as allowed for each position, as determined by the microarray binding assays. The resulting collection of identified molecules was consolidated to eliminate redundant hits, filtrated to identify those including the motif in their intracellular domain and further processed to exclude kinases and phosphatases, since the motif tended to be represented in the active center of such functional domains. The final, curated collection of candidates included 304 type-I transmembrane proteins presenting the motif in the intracellular region.

**Cell lines and reagents.** HEK-293T, HeLa, and THP1 cells were obtained from the American Type Culture Collection. The $ATG16L1^{-/-}$ HCT116 cell line was obtained from Dr. David Boone (Indiana University School of Medicine, South Bend, Indiana, USA)[37], and $Atg16L1^{-/-}$ MEFs were kindly shared by Dr. Ramnik Xavier (Broad Institute MIT and Harvard, USA)[38]. $ATG16L1^{-/-}$ HCT116 cells restored with FL-ATG16L1 or separate Nt (1–299) and Ct (300–607) domains of ATG16L1 were generated by retroviral transduction of the relevant DNA constructs[15]. $Atg16L1^{-/-}$ MEFs were engineered to study IL-10 signaling by transducing them first with a retroviral STAT3-luciferase reporter system and a plasmid expressing human STAT3. Cells were then restored with retroviral constructs expressing FL-ATG16L1 or the N-terminal region (residues 1–299). Finally, cells were transduced with a mixture of retroviral supernatants harboring human IL-10RA-AU and IL-10RB-Flag. An additional strain expressing ectopic AU-LC3B (human) via retroviral transduction was generated to study the colocalization between endocytosed IL-10RB and LC3 in response to IL-10 treatment. THP1 cells were infected with lentiviruses expressing CRISPR/Cas9 guides directed to positions 78–97 or 1025–1044 of the ATG16L1 mRNA (counting from the first coding nucleotide in ATG16L1 mRNA); see below Transfections and retroviral/lentiviral transductions for further details. The latter were then restored via retroviral transduction of a construct expressing the Nt domain of ATG16L1 (1–299) or an irrelevant vector as a control. Cells were cultured at 37 °C and a humidified 5% $CO_2$ atmosphere, in DMEM (Invitrogen) containing 10% heat-inactivated FBS (Invitrogen), 2 mM glutamine and 100 U/ml of penicillin/streptomycin (Invitrogen). Bafilomycin, pepstatin and rapamycin were purchased from Sigma, and E64d from Calbiochem. Recombinant human and mouse IL-10, LIF, and M-CSF were obtained from Life Technologies, R&D, Peprotech and BioLegend, respectively. For pulse-chase experiments in response to IL-10, cells were incubated with the cytokine for 30 min on ice, washed twice and incubated at 37 °C for the relevant times before lysis for Western-blotting or fixation for immunofluorescence.

**DNA constructs.** All constructs were cloned into the pEAK series of mammalian expression vectors or a retroviral derivative called P12-MMP. DNA constructs expressing HA-tagged and GST-tagged human ATG16L1, ΔWDD (1–319) and (1–299), WDD (320–607) and (300–607), AU-LC3, GFP-LC3, GFP-ATG16L1 have

been generated by PCR and fully verified by sequencing[15,17]. Human full-length cDNAs for IL-2Rγ (BC014972), IL-10RB (BC001903), IL-4RA (BC136365), IL-6RB (BC117402), IL-12RB2 (BC104772), IL-20RA (BC113574), IL-21R (BC004348) and IL-31RA (BC110490) were obtained from the Mammalian Gene Collection (MGC, Dharmacon, GE Healthcare). The cDNAs for IL-18R1 and IL-18RAP were purchased from Origene (refs. SC117718 and SC117717, respectively). The IL-17RD cDNA was an incomplete clone purchased from GE healthcare (BC111702; MGC, Dharmacon) where the first native aminoacid is G43 (as in RefSeq NM_017563), thus lacking the N-terminal signal peptide. To allow expression of the clone, the cDNA was fused in-frame to a heterologous leader cassette from the human CD5 protein that leaves an in-frame HA tag at the N-terminus upon cleavage of the leader sequence (CD5L-HA). The cDNA for IL-33R (IL-1RL1; ST2) was kindly shared with us by Dr. Ken Yanagisawa (Jichi Medical School, Tochigi, Japan). The different interleukin receptors were C-terminally tagged with GST, HA or Flag by PCR and verified by sequencing. The sequences of the relevant oligos are shown in Supplementary Table 1. Mutants in critical positions of the WDD-binding motif were introduced by site-directed mutagenesis and verified by sequencing (oligonucleotide sequences shown in Supplementary Table 2). The IL-18R1 cDNA contained a 57 base pair insertion compared with the RefSeq clone NM_003855, likely representing an unremoved intron. This anomalous sequence introduced a premature stop codon that resulted in a truncated protein, and was removed by site-directed mutagenesis (oligonucleotide sequences shown in Supplementary Table 2). The STAT3 cDNA was obtained from Addgene (pLEGFP-WT-STAT3; ref. 71450) and swapped into the P12-MMP retroviral vector. Tagged constructs were generated by PCR and fully verified by sequencing. To reconstitute the IL-10 signaling system in MEFs, the IL-10RB-Flag cDNA was cloned upstream of an IRES-GFP cassette to verify equal transduction efficiency. The IL-10RA cDNA (GE-Dharmacon; BC028082) was C-terminally tagged with AU1 and also cloned into P12-MMP. The retroviral STAT3-luciferase reporter vector was derived from a NF-κB-luciferase retroviral construct originally obtained from Dr. Felix Randow (LMB, Cambridge, UK). The NF-κB enhancer was excised using HpaI/XhoI and an oligonucleotide including 4× STAT3 binding sites was inserted 3 times using the same restriction sites (oligonucleotide sequence shown in Supplementary Table 3).

**Transfections and retroviral/lentiviral transductions.** Transfections were done using the jetPEI lipid reagent (Polyplus) following the instructions provided by the manufacturer. Retroviral transductions were performed to express transgenes cloned into the P12-MMP retroviral vector. Virus-containing supernatants were generated by co-transfecting HEK-293T cells with the relevant P12-MMP constructs together with helper plasmids expressing gag-pol (pMD.gag-pol) and env (VSV-G; pMD-G). Infections were carried out by diluting the viral supernatants with fresh medium (1:1) in the presence of polybrene (8 μg/ml), and spinning the resulting mix onto the target cells for 1 h at 2000 rpm, 32 °C. To deplete ATG16L1 in THP1 cells and Rubicon in MEFs using the CRISPR/Cas9 system, we relied on the lentiCRISPRv2 vector (Addgene, refs. 52961 or 98293). RNA guides were selected using the BreakingCas server at the Centro Nacional de Biotecnología (Madrid, Spain; https://bioinfogp.cnb.csic.es/tools/breakingcas/index.php; oligonucleotide sequences shown in Supplementary Table 4), cloned into this vector using BsmB1 restriction sites and the resulting constructs co-transfected with the helper plasmids psPAX2 (Addgene ref. 12260) and pCMV-VSV-G (Addgene, ref. 8454) into HEK-293T cells. Viral supernatants were collected, filtered and diluted 4:1 with fresh medium. Infections were carried out by spinning the resulting mixture (1 h, 2000 rpm, 32 °C) onto the target cells in the presence of polybrene (8 μg/ml). Cells were selected in puromycin (1 μg/ml) or blasticidin (2 μg/ml) for a week and lysed to evaluate protein expression. The resulting polyclonal population exhibited extensive protein depletion (beyond 95%) and was used as such for further studies.

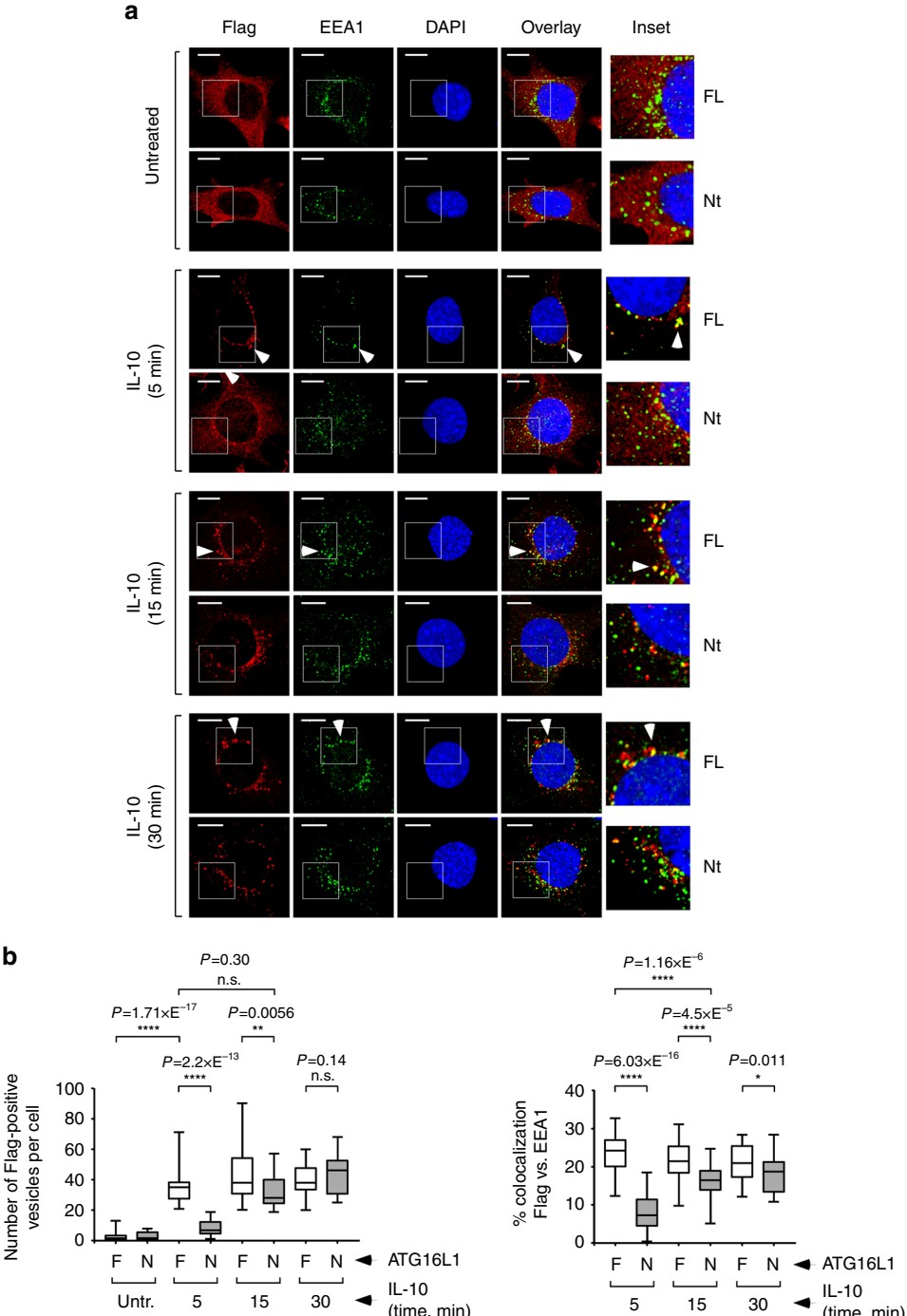

**Fig. 5 Effect of the WDD in the early intracellular trafficking of IL-10/IL-10R endosomes. a** Immunofluorescence assays to explore IL-10-induced IL-10RB trafficking to early endosomes. The indicated cells (FL or Nt) were continuously treated with IL-10 (50 ng/ml) for the shown times and fixed for immunofluorescence with anti-Flag (red) and anti-EEA1 (green) antibodies. Shown are representative confocal pictures. Arrows indicate examples of colocalization events. **b** Quantification of the phenotypes observed in **a**. The left panel shows the number of Flag-positive vesicles per cell at different times after IL-10 treatment in this experiment, and the right panel represents the percentage of colocalizing red (Flag) and green (EEA1) signals per cell (F: FL; N: Nt). Data are presented as box-plots where the central line represents the median value, the box shows percentiles 25–75 and the whiskers include the most extreme values ($n = 27$ cells; n.s: $P > 0,05$; *$P < 0.05$; **$P < 0.01$; ****$P < 0.0001$, two-sided Student's $t$-test).

**Western blotting and co-precipitation studies**. To obtain total lysates for polyacrylamide gel electrophoresis and Western-blotting, cells were collected by centrifugation and resuspended in 2× SDS sample buffer (SB) lacking β-mercaptoethanol and bromophenol blue (100 mM Tris-HCl pH 6.8, 4% SDS, and 20% glycerol) supplemented with PMSF (10 μg/ml, Roche) and a Protease Inhibitor Cocktail (PIC, Sigma). After vigorous vortexing, samples were boiled for 15 min and then spun for 5 min at top speed to remove cellular debris. To generate cytoplasmic and nuclear extracts, cells were lysed on ice for 30 min in a buffer containing 1% Igepal CA-630 detergent (Sigma), 150 mM NaCl, 50 mM Tris pH 7.5, 5 mM EDTA, protease inhibitors (PMSF and PIC) and phosphatase inhibitors (β-glycerophosphate (25 mM, Sigma); sodium fluoride (5 mM, Sigma) and ortho-vanadate (1 mM, New England Biolabs)). After a 10 min centrifugation step, the supernatant was collected as a cytoplasmic lysate. To obtain the corresponding nuclear extracts, the remaining pellet was lysed in SB as described above. Protein concentrations were measured using the DC Protein Assay (BioRad). For co-immunoprecipitation studies, cells were lysed in the same 1% Igepal CA-630 buffer

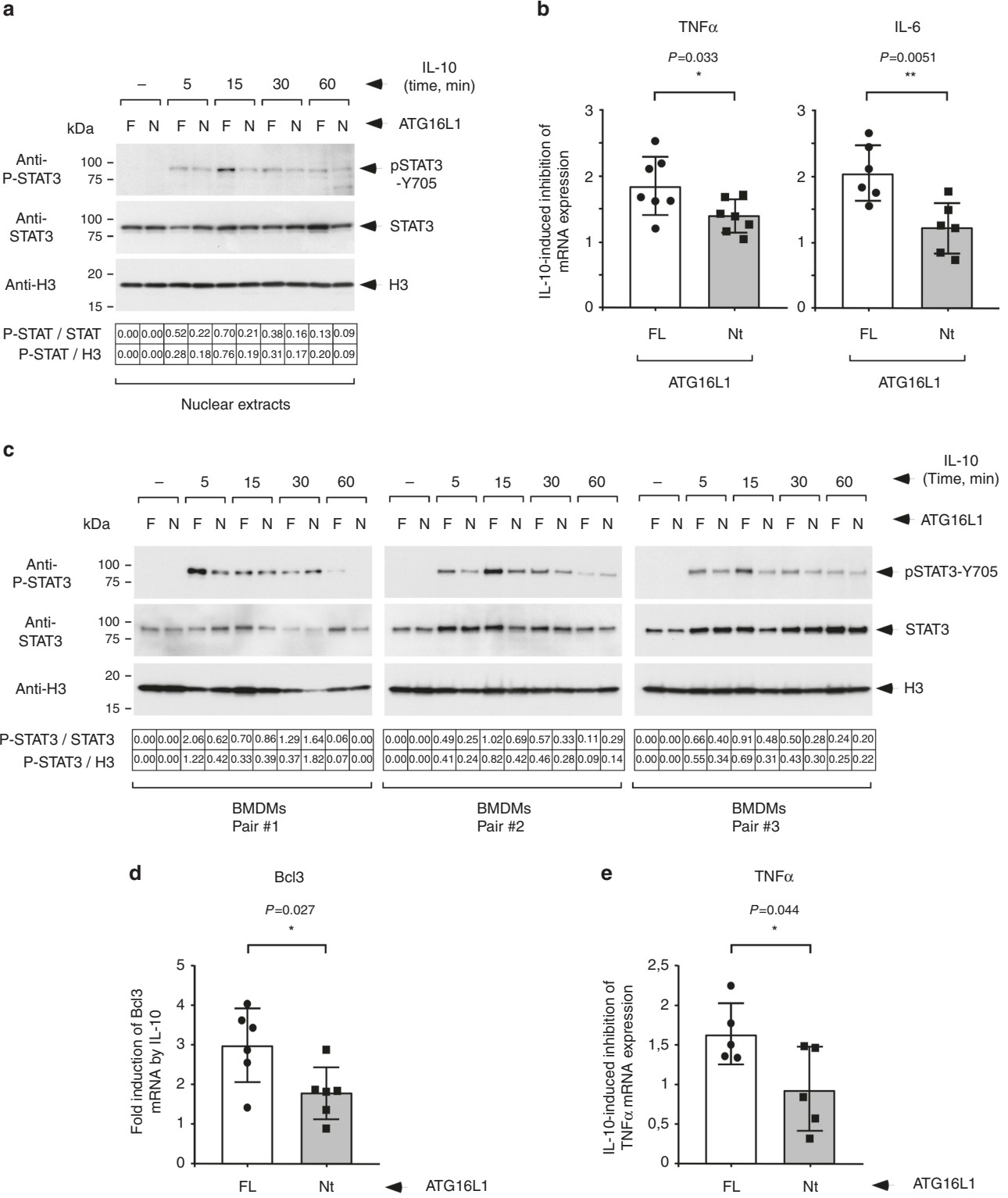

described above and the resulting cytoplasmic lysates diluted 1:5 in the same buffer lacking detergent to a final detergent concentration of 0,2% to preserve protein interactions. Lysates were pre-cleared with agarose beads (1 h, 4 °C, rotation) and then subjected to immunoprecipitation with the relevant antibodies (2–3 h, 4 °C, rotation) plus agarose beads coupled to protein G (1 h, 4 °C, rotation), or agarose beads coupled to GSH (2–3 h, 4 °C, rotation). Immunoprecipitating antibodies: Anti-Flag (1/150, D6W5B, rabbit mAb, Cell Signaling 2368), anti-IL-10RB (1/50, F6, mouse mAb, Santa Cruz sc-271969), anti-ATG16L1 (1/100, rabbit polyclonal, MBL PM-040), irrelevant rabbit immunoglobulin (Merck, Millipore). Beads were washed 3–4 times with ice-cold immunoprecipitation buffer (0,2% detergent), resuspended in 2× Reducing Sample buffer (SB, supplemented with

β-mercaptoethanol (10%) and bromophenol blue (0,2%)), and boiled for 10 min. Equal amounts of protein or immunoprecipitated eluates were resolved by SDS-PAGE, transferred to a polyvinilidene difluoride membrane (Millipore) to be probed with the relevant primary antibodies: anti-GST (1/1000, rabbit polyclonal, Cell Signaling 2622), anti-HA (1/1000, 16B12, mouse mAb, Babco BioLegend 901501), anti-AU1 (1/1000, rabbit polyclonal, Thermo PA1-26548), anti-Flag (1/1000, M2, mouse mAb, Sigma F3165), anti-GFP (1/5000, B34, mouse mAb, BioLegend 902601), anti-ATG16L1 (1/2000, 1F12, mouse mAb, MBL M150-3), anti-IL-10RA (1/500, A3, mouse mAb, Santa Cruz sc-365374), anti-IL-10RB (1/500, F6, mouse mAb, Santa Cruz sc-271969), anti-phospho-STAT3-Y705 (1/1000, rabbit polyclonal, Cell Signaling 9131), anti-STAT3 (1/1000, mouse mAb, Cell Signaling

**Fig. 6 Function of the WDD in IL-10 signaling in THP1 cells and BMDMs. a** Pulse-chase assay of IL-10-induced STAT3 phosphorylation in differentiated THP1 cells. The indicated strains (F: endogenous full-length ATG16L1; N: CRISPR/Cas9-induced ATG16L1 depletion (guide 1025–1044) plus restoration with the Nt domain, aminoacids 1–299) were treated with PMA (125 ng/ml; 48 h), pre-incubated with IL-10 (50 ng/ml; 30 min on ice), incubated at 37 °C for the shown times and lysed to obtain nuclear extracts for the indicated Western-blots. Corrected densitometric quantifications are shown at the bottom of the Western-blot images. **b** Quantitative PCR to assess IL-10-induced suppression of TNF and IL-6 mRNA upregulation by LPS in THP1 cells. Cells were pre-activated as in **a**, treated with LPS (TNF, 10 ng/ml; IL-6, 10 or 60 ng/ml) for 5 h with or without IL-10 (50 ng/ml) and lysed for qPCR. Shown are average fold inhibition values induced by IL-10 −/+ s.d. (TNF: $n = 7$, IL-6: $n = 6$ experimental points; *$P < 0.05$; **$P < 0.01$, two-sided Student's $t$-test). **c** Pulse-chase assay of IL-10-induced STAT3 phosphorylation in BMDMs. BMDMs from 3 randomly-chosen mouse pairs (F: full-length ATG16L1; N: E230 mice expressing the Nt domain) were pre-incubated with IL-10 (50 ng/ml; 30 min on ice), incubated at 37 °C for the shown times and lysed to obtain nuclear extracts for the indicated Western-blots. Corrected densitometric quantifications are shown at the bottom of the Western-blot images. A statistical analysis of nuclear P-STAT3 signals is shown in Supplementary Fig. 16c. **d** Quantitative PCR to evaluate IL-10-induced expression of Bcl3. BMDMs from the indicated mice were treated with IL-10 (50 ng/ml, 4 h) and lysed for qPCR. Shown are average fold induction values induced by IL-10 −/+ s.d. ($n = 6$ mice; *$P < 0.05$, two-sided Student's $t$-test). **e** Quantitative PCR to assess IL-10-induced suppression of TNF mRNA upregulation by LPS. BMDMs from the indicated mice were treated with LPS (1 ng/ml; 6 h) with or without IL-10 (50 ng/ml) and lysed for qPCR. Shown are average fold inhibition values induced by IL-10 −/+ s.d. ($n = 5$ mice; *$P < 0.05$, two-sided Student's $t$-test).

9139), anti-IL-10 (1/1000, D13A11, rabbit mAb, Cell Signaling 12163), anti-Histone-3 (1/1000, D1H2, rabbit mAb, Cell Signaling 4499), anti-LC3 (1/1000, mouse mAb, MBL M186-3), anti-p62 (THPs, 1/1000, mouse mAb, BD 610833; MEFs 1/1000, rabbit polyclonal, MBL PM045), anti-Rubicon (1/1000, D9F7, rabbit mAb, Cell Signaling 8465), anti-GAPDH (1/10000, 6C5, mouse mAb, Abcam ab8245). After incubation with the appropriate secondary HRP- (Jackson Immunoresearch) or DyLight- (ThermoFisher) coupled antibodies (1/10000), blots were developed by chemiluminescence using the ECL (Amersham) or Odissey infrared imaging systems, respectively. Signal quantifications were carried out with the ImageJ software. Crude densitometric values for P-STAT3 or IL-10 were corrected by the values provided by control Western-blot signals (total STAT3, GAPDH or Histone-3, H3).

**Flow cytometry.** Cells were detached using the trypsin-free Accutase Solution (Sigma), resuspended in culture medium containing 0,1% azide and a 1:100 dilution of mouse or rat immunoglobulins (Sigma) as blocking agents, and incubated for 30 min on ice. The specific antibodies were then added to the cell suspension at a 1:100 dilution, and an additional incubation was carried out on ice for 30 min. Cells were then washed repeatedly. All specific antibodies were purchased from BD: anti-mouse-CD11b-BB515 (rat, BD 564454), anti-mouse-F4/80-Alexa Fluor 647 (rat, BD 565853), anti-human-IL-10RA-Alexa Fluor 647 (rat, BD565255), anti-human-IL-10RB-Alexa Fluor 647 (mouse, BD 564372). Control antibodies were: mouse IgG1-Alexa Fluor 647 (BD 557714) and rat IgG2b-BB515 (BD 564421). Cells were analyzed in an Accuri C6 (BD BioSciences) cytometer running on version 1.0.264.21 of the Accuri C6 software. Details on the gating and quantification strategies are provided in the figure legends and Supplementary Fig. 17.

**Immunofluorescence and confocal microscopy.** Cells were seeded onto poly-L-lysine coated coverslips, transfected or treated the next day and, in most cases, fixed in 4% paraformaldehyde (RT, 15 min). Preparations were permeabilized in a PBS/0,5% Igepal/100 mM glycine solution (RT, 30 min), blocked in PBS/3% BSA (RT, 45 min) and stained with primary and secondary antibodies (RT, 1 h) diluted in PBS/2%BSA. Samples generated to detect LC3 with an anti-LC3 antibody were fixed with methanol (100%, −20 °C, 5 min) and further processed as described above. Primary antibodies were: anti-Flag (in most cases rabbit polyclonal (1/2000, Cell Signaling 2368); mouse monoclonal (1/100, Sigma F3165) for co-staining with anti-EEA1), anti-EEA1 (1/750, EPR4245, rabbit mAb, Abcam ab109110), anti-LC3 (1/200, 5F10, mouse mAb, NanoTools 0231-100), anti-LAMP1 (1/200, EPR21026, rabbit mAb, Abcam, ab208943), anti-Rab7 (1/100, EPR7589, rabbit mAb, Abcam, ab137029). Secondary antibodies were AffiniPure goat anti-mouse or anti-rabbit-Cy3 and goat anti-mouse or anti-rabbit Cy5 (Jackson Immunoresearch, 1/400). Studies with Dextran as an endocytic tracker (Dextran-Alexa Fluor 647, 10.000 MW fixable, Invitrogen, D22914) were done by co-incubating the relevant cell lines with IL-10 (40 ng/ml) and dextran (100 µg/ml) for the required times and then processing them following the same protocol used for conventional immunofluorescence, avoiding the detergent permeabilization step. Analysis of surface IL-10R expression by immunofluorescence was done by incubating the adhered cells with the same primary antibodies and under the same conditions as for flow cytometry (see above), and subsequently fixing them in 4% paraformaldehyde as for immunofluorescence. Samples were analyzed in a Leica SP5 confocal microscope. Except for surface IL-10Rs, images obtained for Cy5 were pseudocoloured in green for more clarity. Scale bars represent 10 µm in all micrographs. Colocalizations were evaluated in transfected HeLa cells by blindly scoring the percentage of receptor-positive vesicles (stained with anti-Flag and Cy3) that show a clearly overlapping signal with GFP (GFP-LC3 or GFP-ATG16L1) in the confocal pictures. Colocalization between endocytosed IL-10R and EEA1, LAMP1, Rab7, or LC3 was evaluated using the Colocalization Statistics software of the LAS AF X Core Suite, version 2.7.3.9723 (Leica Microsystems). The number of IL-10RB-

positive vesicles per cell was established by blindly counting the labeled vacuoles on confocal images.

**Luciferase assays.** Engineered MEFs were plated in complete medium, pre-incubated the next day for 4 h in medium containing 0,5% FCS, treated with the cytokine in the same low serum medium and lysed for luciferase activity using the Luciferase Assay System (Promega), essentially following the instructions provided by the manufacturer.

**Assays with THP1 cells and BMDMs.** THP1 cells were pre-treated with PMA (125 ng/ml) for 48 h in complete medium (10% FCS) to promote their differentiation to a macrophage-like phenotype. Cells were then incubated in 1% FCS for 5 h before treatment to evaluate IL-10 activity (Western-blot and qPCR). Wild-type and E230 mice[24] were sacrificed using a $CO_2$ chamber and immediately dissected to isolate bone marrow cells by flushing the tibias and femurs with a 23G syringe filled with cold complete culture medium containing 20% FCS. Cells were washed, plated at a density of approximately $2 \times 10^6$ cells/ml and treated with 20 ng/ml M-CSF for 6 days, with a medium change at day 3. Adherent cells were then harvested using a cell scraper and plated in complete medium containing 20% FCS and lacking M-CSF for 16 h. Cells were incubated the next day for 5 h in DMEM/1% FCS before conducting different assays in response to IL-10 (Western-blot and qPCR). Depending on the final cell yield, different number of experimental replicas were used for each WT/E230 mouse pair.

**Quantitative PCR.** THP1 cells and BMDMs were washed and lysed in-plate using the RNeasy Mini Kit (Qiagen) for RNA isolation. Quantitative PCR was carried out using the qPCR iTaq Universal SYBR Green One-Step kit (BioRad) using 30–50 ng of total RNA as a template. Oligonucleotides were selected using the PrimerBank Primer Search on-line tool (MGH, Harvard; https://pga.mgh.harvard.edu/primerbank/); sequences are shown in Supplementary Table 5. Data were collected using the BioRad iQ5TM 2.0 Standard Edition Optical System Software, and later corrected against the results obtained for housekeeping genes (Actin, GAPDH) following the $2^{-\Delta\Delta CT}$ method[39].

**Animals and ethics.** Mice were maintained in ventilated racks in a Specific Pathogen-Free facility under controlled temperature (23 °C), humidity (50%) and light/dark cycle (12 h/12 h) conditions, strictly following European Union regulations. All procedures were evaluated and approved by the Ethics and Animal Wellbeing Committees of the University of Salamanca and the Junta de Castilla y Leon local government.

**Statistics and reproducibility.** Unless otherwise stated, all graphs represent mean −/+ standard deviation values obtained from the indicated number of experimental points (n). Graphs were generated using GraphPad Prism 7.04, except those in Fig. 1b, c and Supplementary Fig. 1a that were created with Excel 1902. Box-plots were used to represent the number of intracellular vesicles per cell and the degree of colocalization with endocytic markers in those figures displaying intracellular trafficking data obtained from engineered MEFs, since this plot type displays more graphically the wide experimental dispersion that we found in these assays. In the shown box-plots, the box indicates numerical data included in percentiles 25–75, the central line represents the median value and the whiskers include the most extreme values. Statistical significance was established in all cases by calculating the $P$-values resulting from two-tailed Student's $t$-tests. All experiments were reproduced at least 3 times with similar results.

**Reporting summary**. Further information on research design is available in the Nature Research Reporting Summary linked to this article.

## Data availability

The database of human type-I transmembrane proteins used to search for the WDD-binding motif is available as Supplementary Data 5. Information about how to retrieve this database from the public Uniprot sequence bank is provided in the "Methods" section (under Prosite Comparisons). All data are available in the article and Supplementary files or from the corresponding author upon reasonable request. Source data are provided with this paper.

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

## Acknowledgements

We thank Dr. Ken Yanagisawa (Jichi Medical School, Tochigi, Japan) for kindly sharing with us a plasmid expressing the IL-33R, Alberto Ruiz for help with the management of Excel tables, and members of the CIC for support. This work was funded by grants from the Ministerio de Ciencia e Innovación of the Spanish Government (refs. SAF2014-53320-R and SAF2017-88390-R), the Junta de Castilla y León local government (ref. SA042P17), the Broad Medical Research Program (Crohn's and Colitis Foundation of America, CCFA; ref. IBD-0369) and the Fundación Solórzano (ref. FS/18-2014). The Centro de Investigación del Cáncer is supported by the Programa de Apoyo a Planes Estratégicos de Investigación de Estructuras de Investigación de Excelencia cofunded by the Castilla y León autonomous government and the European Regional Development Fund (CLC-2017–01). Additional funding comes from the Fondo Europeo de Desarrollo Regional (FEDER) program of the European Union. This work was funded in part by Biotechnology and Biological Sciences Research Council (BBSRC) grant BB/R00904X/1 to TW, U.M., S.C., and P.P.P. and through BBSRC Institute Strategic Programme Gut Microbes and Health BB/R012490/1: BBS/E/F/000PR10353, BBS/E/F/000PR10355. I.S.G. was supported by predoctoral fellowships from the Fundación Moraza and the Estrategia Regional de Investigación e Innovación (Junta de Castilla y León and FEDER; ref. CLC-2017-01). E.B. was funded by a predoctoral fellowship from the FPU program (Ministerio de Educación, Spanish Government). R.V. holds a pre-doctoral contract co-financed by the Junta de Castilla y León and the Fondo Social Europeo (European Union). A.F. is supported by a predoctoral fellowship from the Universidad de Salamanca. F.X.P. holds a tenured position at the CSIC.

## Author contributions

I.S.G., E.B.R., R.V., A.F.C., J.L.C., and A.M.R. performed the experiments. U.M., S.C. and T.W. designed and generated E230 mice lacking WDD. U.M. designed genotyping protocols. Loss of WDD function in adult mice was verified by P.P.P. and T.W. P.P.P. and T.W. provided critical reading of the manuscript. I.G.H. contributed funding and reagents. F.X.P. conceived and designed the study, and wrote the manuscript.

## Competing interests

The authors declare no competing interests.
