## [Peer Review File · Nature Communications]

Reviewers' comments:

Reviewer #1 (Remarks to the Author):

Previous research by the investigator's group showed a specific for the WD40 domain of ATG16L1 different from its contribution to canonical autophagy and identified TMEM59 as a ATG16L1 WD40 interacting partner to induce LC3 lipidation targeted to the endosomal compartment. In this new study, they have advance that discovery to, using a peptide library, define a more accurate consensus for the WD40 target sequence and identify putative novel proteins that may interact with ATG16L1 through its WDD. Interestingly, several cytokine receptors appear to possess those sequences, opening the exciting possibility for a non-canonical autophagy role of ATG16L1 in the regulation of cytokine receptor trafficking and function. For two of these receptors, the IL-2Rgamma chain and the IL-10RB, this manuscript shows compelling evidence that supports their interaction with the ATG16L1 WDD and the ability of this interaction to led to the recruitment of LC3 to the endosomal vesicles containing those receptors. Finally, using different biological systems, the authors generate very interesting data that support a positive role of this interaction for IL-10R signaling.

One of the most important issues, however, that remain unresolved in this manuscript is how the biochemical data shown in the first part of the manuscript may translate into the functional consequences observed on IL10R signaling. Though the authors proposed the possibility that defective endosomal generation may affect signaling, as these vesicles have been shown to be act for several receptors as signaling platforms, the data presented appear to show somehow data that may not fit with that theory. For instance, clear LC3 flux induce by the interaction of ATG16L1 and the IL-10R and the possibility that these endosomes end up fusing with autophagosomes to form amphisomes, all point to a degradative nature of the role of LC3 activation by ATG16L in the IL-10R induced endosomes. The increased intracellular IL-10 levels (though no data is offered on the actual IL-10R levels) in the presence of FL Atg16L1 vs N-t fragment would however point in the opposite direction. This mechanism requires further exploration to understand the consequences of this process (decreased degradation, trafficking to a different compartment...) that can account for the increased signaling cause by the interaction of the IL-10R with the ATG16L1 WDD. It would also be important to show that altered signaling is not due to effects other than on the IL-10R of ATG16L1 that may indirectly affect IL-10R trafficking, degradation or function. This would be easily demonstrated in the MEF reconstituted system by comparing signaling strength of the wt and the 5M mutant IL-10RB in the presence of FL or Nt ATG16L1.

Whereas the data on Figs 4-5 shows that, as likely expected, the interaction of the IL-10R with ATG16L1 is regulated by ligand binding to the receptor, the data on transfected HEK cells shows this interaction occurring in cells expressing single chains, instead of full receptors, in the absence of any ligand. Similarly, the generation IL-10Rb or IL-2Rgamma containing endosomes occurs in the absence of any ligand or complete receptors. This could be a consequence of the levels of overexpression achieved in transfected HEK cells, but if this is the case the functional relevance of the data would be quite limited compare to that presented in the MEF system.

The claim that this process is different from the recently described LANDO, is based on differences in the kinetics of both processes. However, the recently described LANDO has been shown in a completely different system, and comparing kinetics in different cell types in response to different stimuli might be not the best way to assess whether these two processes are indeed different. More clear cut data supporting differences would be required to allow that statement (e.g. is the Rubicon-dependency different between the process described in this manuscript and LANDO?) Whereas it is easy to understand why STAT3 (total) and pSTAT3 (cytosolic ready to be imported into the nucleus) levels should be different in the immunoblots on cytosolic extracts, it is not clear why this is also a case when using nuclear extracts, as almost all STAT3 detected in the nucleus should be pSTAT3. It would also help that in those immunoblots that show differences in signaling (Figs4 and 5), quantifications of immunoblots and statistical analysis would be offered to better appreciate the significance of the differences in signaling events in the different situations analyzed.

Reviewer #2 (Remarks to the Author):

In this manuscript, Serramito-Gomez and colleagues report that the WD40 domain of ATG16L1, a key protein in LC3 lipidation during autophagic process, is required for IL-R signalling modulation at endosomes. Overall, the study is well addressed and presented data are sustained by robust biochemical analyzes. However, the ins and outs and the rational of the study are not always very clearly presented and the manuscript is sometimes experimentally complicated. With a precise and dedicated revision, the paper could be of interest for readers of Nature Communications.

MAJOR CONCERNS:

- My major remark mostly concerns experimental emphasis of the paper and final message of the authors. While a considerable amount of efforts have been made in the biochemical description of the screen that lead to IL-receptors identification and in the importance of ATG16L1 WD40 domain, the final conclusion suggests that IL-10RB and IL-2Rgamma endo-lysosomal trafficking is a key target of IL-R/ATG16L1 interaction relevancy. This "dynamic and endosomal trafficking" hypothesis could thus benefit of additional experimental framework to propose a more detailed mechanistic explanation. While experiments using EEA1 staining are informative, this part of the message could be strengthened by dynamical analyzes of "in time/in space" IL10 endocytosis and sorting with other techniques, such as fluorescent dextran pulse-chase experiments and other endo-lysosomal markers.

MINOR COMMENTS:

- 1) While I found that the paper is very well written, I think that it could benefit from a more detailed discussion section, considering the important amount of data to be discussed.
- 2) In line with this – and this is only a suggestion – the title could mention "endosomal cytokine signalling" rather than simply "cytokine signalling".
- 3) The section about "autophagic activity" of IL-2R, IL-10R and IL-33R is not very clear.
- 4) Could the authors precise with more details why they focused on IL-10RB associated signalling in this context?
- 5) It is not clear to me why the LC3 WB shown in Sup Fig5b only displays one band?
- 6) The increased levels of IL-R in ATG16L1 FL or Nt cells shown by western blot in Sup Figure 5c could benefit of quantification(s).
- 7) Surface exposure analyzes of IL-10R were analyzed by flow cytometry. Is that possible to show similar experiments in immunofluorescence?
- 8) As LC3 is not only present on (transient) autophagosome, but also on pre-autophagosomal structures (omegasomes, phagophores) as well as autolysosomes, the notion of "amphisome" which is proposed by authors (Sup Figure 7) is not very convincing in this context. Amphisomes existence is still under debate and thus needs, at a minimal, a full characterization using biochemical and morphological precise techniques.

In the following we address the specific comments of the reviewers.

REVIEWER #1

We appreciate the reviewer's careful reading of the manuscript and the suggestions. We are providing a point-by-point reply.

POINT 1. Reviewer #1 mentioned that “One of the most important issues, however, that remain unresolved in this manuscript is how the biochemical data shown in the first part of the manuscript may translate into the functional consequences observed on IL10R signaling. Though the authors proposed the possibility that defective endosomal generation may affect signaling, as these vesicles have been shown to be act for several receptors as signaling platforms, the data presented appear to show somehow data that may not fit with that theory. For instance, clear LC3 flux induce by the interaction of ATG16L1 and the IL-10R and the possibility that these endosomes end up fusing with autophagosomes to form amphisomes, all point to a degradative nature of the role of LC3 activation by ATG16L in the IL-10R induced endosomes. The increased intracellular IL-10 levels (though no data is offered on the actual IL-10R levels) in the presence of Fl Atg16L1 vs N-t fragment would however point in the opposite direction. This mechanism requires further exploration to understand the consequences of this process (decreased degradation, trafficking to a different compartment...) that can account for the increased signaling cause by the interaction of the IL-10R with the ATG16L1 WDD”.

We agree this is a critical point. We have addressed this issue experimentally from a variety of angles.

Before describing the new results, we believe it is important to point out that the set of biochemical studies presented in the first part of the manuscript was mainly designed as a simple and quick means to select suitable candidate receptors more likely to present WDD-dependent features, not to draw detailed mechanistic information about the role of the WDD in the physiology of the receptors in response to their natural ligands. Such preliminary selection procedure allowed us to identify candidate receptors able to interact with the WDD in a biological system (co-immunoprecipitation experiments), and to establish that this interaction is truly mediated by the WDD-binding motif and is able to activate the LC3-lipidation capabilities of the LC3 lipidation complex. Since all these experiments are based on overexpression of the receptors in the absence of their receptor partners or ligand stimulation, they are not informative regarding the functional consequences that their binding to ATG16L1 through the WDD might have in their normal physiology in response to ligand stimulation. Therefore, the physiological impact of the WDD-dependent mechanism that we describe in our paper should be concluded only from experiments performed in reconstituted MEFs and THP1/BMDM cells, and detailed mechanistic information regarding the endocytosis and trafficking features of engaged IL-10Rs arises exclusively from the MEF system. Therefore, we believe that no conclusions in this regard are to be drawn from the overexpression studies. These ideas divide our manuscript in two differentiated parts and, in our view, were not sufficiently explained in our previous manuscript version. We hope they are now explained more clearly in a new text included in **page 8** of the revised manuscript.

As requested by the reviewer, in order to “understand the consequences of this process (decreased degradation, trafficking to a different compartment...) that can account for the increased signaling” we have extended our intracellular trafficking studies in MEFs to additional subcellular markers (LAMP1 and Rab7), pulse-chase experiments (for EEA1 and

LAMP1) and IL-10R trafficking experiments involving inhibition of lysosomal degradation with bafilomycin A1.

First, we found that the level of colocalization between endocytosed IL-10RB and the early endosomal marker EEA1 follows in the new pulse-chase assays the same trend observed in the continuous IL-10 treatment experiment originally shown in our initial submission (**old Fig. 4d, now included in an independent new Fig. 5 in the revised version**). That is, we detected again a significantly reduced rate of IL-10RB-Flag/EEA1 colocalization in Nt cells early after IL-10 activation (5 and 15 min time points), but also when comparing experimental points with similar numbers of IL-10RB-Flag-positive vesicles (FL 5 min vs Nt 15 min). These new pulse-chase results are shown in a new **Suppl. Fig. 10**, and are briefly described in **page 11** of the revised manuscript. On the other hand, FL and Nt cells showed similar colocalization rates between IL-10RB and LAMP1 (both in continuous IL-10 treatment and pulse-chase studies) or Rab7 (only continuous treatment), and both in the 15 min time point and also when comparing FL 5 min with Nt 15 min data. We only detected decreased colocalization rates of these late endosome markers with IL-10RB in Nt cells in the 5 min post-IL-10 treatment point, where there are almost no IL-10RB-positive endosomes in Nt cells and therefore this parameter is probably irrelevant. These new data are shown in **Suppl. Figs. 11-13** and briefly described in **page 11 (bottom)** of the revised text.

In our view, these results show that absence of the WDD causes a defect in the trafficking of IL-10/IL-10R complexes to early endosomes, whereas their targeting to late endosomal/lysosomal compartments remains for the most part unaffected. Consequently, these results are consistent with the idea that the IL-10/IL-10R endosomes are not degraded at a faster pace in Nt cells, because IL-10R trafficking to the lysosomes (LAMP1-positive signal) is not favoured in these cells. Therefore, enhanced IL-10R endosome degradation does not seem to be the main alteration that explains why the number of IL-10R-positive endosomes is diminished in cells lacking the WDD. Interestingly, these data also indicate that IL-10R endosomes do not access the lysosomal compartment more efficiently in FL cells either, where they are however more prominently labelled with LC3. Consequently, these results also appear to argue against a possible degradative role of LC3 in this context (see below for additional considerations about this point). More generally, the data indicate that endocytosed IL-10Rs reach the same intracellular compartments in both cellular strains, thus pointing to the idea that absence of the WDD does not cause trafficking of the engaged IL10Rs to substantially different subcellular compartments, a question that the reviewer specifically asks. This is, of course, considering the range of markers that we have used in our study, which does not allow us to formally exclude the possibility of a differential trafficking to other unidentified compartments. More systematic studies with a wider range of subcellular markers would be required to answer this question, albeit with a substantial risk since the nature of such possible compartments is currently unknown.

Second, in order to explore in a different way whether the decreased number of IL-10RB-Flag-positive vesicles in Nt cells comes from reduced endocytosis of activated IL-10Rs or increased degradation of IL-10R-positive endosomes, we conducted a sort of “IL-10R endocytic flux” assay where we stimulated the system with IL-10 (in a pulse-chase modality) in the absence or presence of bafilomycin A1 to inhibit lysosomal activity, and then we evaluated the number of intracellular IL-10RB-Flag-positive vesicles per cell in the different conditions. We reasoned that inhibition of lysosomal degradation by bafilomycin would eliminate any contribution of a possibly dissimilar degradation rate of IL-10R endosomes in FL and Nt cells, thus revealing the real contribution of endosome generation to the whole phenotype. In these experiments we found that bafilomycin does not equalize the different number of IL-10RB-Flag-positive vesicles observed in FL and Nt cells, indicating that such difference likely arises for the most part from a reduced rate of endocytosis and/or endosome

formation in Nt cells. This result is consistent with the observation that IL-10Rs do not show enhanced colocalization with LAMP1 in Nt cells (as mentioned above). Importantly, bafilomycin increased the number of Flag-positive vesicles in all conditions, indicating that some degree of IL-10R endosome degradation is taking place. The new bafilomycin experiments are shown in **Supp. Fig. 9b** and described in **page 11 (top)** of the revised text.

Therefore, in response to the specific request of the reviewer that we try to “understand the consequences of this process (decreased degradation, trafficking to a different compartment...) that can account for the increased signaling”, we believe that the main role of the WDD in IL-10R biology is to favour endocytosis and early trafficking of IL10/IL-10R endosomes, without having a major impact on their lysosomal degradation (“decreased degradation”) or in the nature of the endocytic route they follow on their way to the lysosome (“trafficking to a different compartment...”). Still, as mentioned above, we cannot exclude that the WDD could promote their transit to unidentified subcellular compartments that might enable optimal signalling. Such increased endocytosis and early trafficking (in the absence of a dissimilar degradation rate) causes enhanced accumulation of endosomes harbouring activated IL-10Rs in FL cells, which, considering the view that endosomes constitute important signalling platforms, likely results in improved IL-10 signalling output. Reversing the argument, absence of the WDD results in a substantial reduction in the total number of endosomes harbouring activated IL-10/IL-10R complexes mainly by reducing their endocytic input into the pathway, thus causing the impaired IL-10 signalling output that we observe in Nt cells.

How ATG16L1 may promote endocytosis of IL-10Rs is currently unclear. A previously described interaction between the N-terminal region of ATG16L1 and the clathrin heavy chain (Ravikumar et al, 2010) could promote endocytosis upon recruitment of ATG16L1 to the activated IL-10Rs, but a possible role of LC3 lipidation in this scenario would not be obvious. In this context ATG16L1 might function mainly as a scaffold protein, independently of its LC3 lipidation ability. How ATG16L1 may favour efficient early trafficking of IL-10R endosomes is also unclear but, interestingly, in addition to the degradative role of LC3 in a variety of pathways (ranging from canonical autophagy (Bento et al, 2016) to LAP (Sanjuan et al, 2007)), LC3 is also known to favour membrane fusion events (Weidberg et al, 2011), including, for example, those involved in the secretion of lysosomal contents through the osteoclast ruffled border (DeSelm et al, 2011). A role of LC3 in membrane fusion could facilitate the transition of LC3-labelled IL-10R endosomes to EEA1-positive vesicles, a scenario that would point again to a non-degradative function of LC3 in the process that we describe in our study. Such atypical non-degradative function of LC3 may be implicit in our model, since the canonical degradative role of LC3 would be expected to increase fusion of LC3-labelled IL-10R endosomes with the lysosomal compartment to cause earlier signal cessation, instead of an enhanced output. Exploration of these issues, and whether or not LC3 is truly involved in the mechanism that we describe in this manuscript, are certainly very interesting themes that we will address in subsequent studies. More generally, our results underscore the notion that timely endocytosis and proper endocytic trafficking are important factors that maximize the overall signalling output of cytokine receptors. All these considerations are mentioned in **pages 15-16** of a substantially expanded Discussion section now included in the revised manuscript.

The reviewer also mentions in this point that “no [trafficking] data is offered on the actual IL-10R levels”. Before our initial manuscript submission we had carried out a number of Western-blot studies to examine IL-10RB degradation in response to IL-10, but changes IL-10R signal in this experimental setting were difficult to observe. Our interpretation was that a portion of the IL-10RB (Flag) signal was originated from a receptor pool that may not be accessible to IL-10 (perhaps located in the endoplasmic reticulum on its way to the plasma

membrane), or perhaps from a possible remaining fraction of surface IL-10RB that remained unengaged by IL-10. Such IL-10-insensitive IL-10RB signal likely quenches any changes in the total IL-10R levels occurring during receptor intracellular trafficking in FL and Nt cells after IL-10 treatment. This bias does not occur in immunofluorescence studies where the IL-10R-Flag signal is spatially segregated so that we can focus exclusively on intracellular vesicles. Such bias does not occur either if IL-10R trafficking is indirectly followed by blotting for IL-10, as observed in the experiments that we eventually decided to include in **Fig. 4c**.

For a more linear argumentation, in the revised manuscript we have segregated the figures showing just vesicle number data (**now in a new Fig. 4d**) from those showing colocalization assays with EEA1 (**now in an independent new Fig. 5**), which were originally shown together in the context of the colocalization studies with EEA1 (**old Fig. 4d**). This reorganization resulted in one additional main figure showing the EEA1 trafficking data (**new Fig. 5**). In our view, this slightly modified structure allows easier accommodation of the new bafilomycin (**Suppl. Fig. 9b**) and marker colocalization (**Suppl. Figs. 10-13**) data in the overarching logic of the manuscript. Finally, in response to the comments by other reviewer, we have eliminated all references to amphisome formation throughout the manuscript.

POINT 2. The reviewer thinks that *“It would also be important to show that altered signaling is not due to effects other than on the IL-10R of ATG16L1 that may indirectly affect IL-10R trafficking, degradation or function. This would be easily demonstrated in the MEF reconstituted system by comparing signaling strength of the wt and the 5M mutant IL-10RB in the presence of FL or Nt ATG16L1”*.

We agree. This has been done.

As suggested by the reviewer, we retrovirally expressed WT and 5M versions of IL-10RB (along with IL-10RA) in MEFs harbouring FL and Nt forms of ATG16L1. While the expression levels of both versions of IL-10RB (WT and 5M) was comparable in FL cells, unexpectedly, the mutant receptor was substantially overexpressed in comparison with its wild-type counterpart in Nt cells (up to 10 fold; not shown). The reason for this surprising effect is unclear. In addition, the behaviour of these Nt cell variants was erratic in STAT3-luciferase signalling assays in response to IL-10, perhaps due to the imbalanced expression levels of the 5M receptor. We therefore disregarded the Nt strains and focused on cells expressing FL ATG16L1. Signalling experiments indicate that IL-10-induced STAT3 luciferase activity is reduced in cells expressing the 5M mutant by about the same magnitude as observed in cells expressing Nt and unaltered IL-10Rs (**as shown in Fig. 4b**). These results indicate that IL-10RB motif mutation and absence of the WDD cause the same signalling defect.

From these assays we conclude that the defective signalling output caused by absence of the WDD is likely due to the absence of IL-10RB/ATG16L1 interaction, and not to alternative effects that might *“indirectly affect IL-10R trafficking, degradation or function”*, as the reviewer points out. These new results are shown in a new **Suppl. Fig. 14b** and are briefly described in **page 12 (bottom)** of the revised manuscript.

POINT 3. The reviewer mentioned that *“Whereas the data on Figs 4-5 shows that, as likely expected, the interaction of the IL-10R with ATG16L1 is regulated by ligand binding to the receptor, the data on transfected HEK cells shows this interaction occurring in cells expressing single chains, instead of full receptors, in the absence of any ligand. Similarly, the generation IL-10Rb or IL-2Rgamma containing endosomes occurs in the absence of any*

ligand or complete receptors. This could be a consequence of the levels of overexpression achieved in transfected HEK cells, but if this is the case the functional relevance of the data would be quite limited compare to that presented in the MEF system”.

We completely agree. This has been clarified and further explained in the revised manuscript.

As mentioned in Point 1 above, the main intention of the data obtained from the individual receptor overexpression studies that we presented in Figs. 2 and 3 was to follow an elementary procedure to quickly identify those receptors showing WDD-dependent features, so that we could focus on one of them to eventually provide proof-of-principle that the WDD regulates receptor biology. As the reviewer points out, all parameters tested in Figs. 2 and 3 are the consequence of the massive receptor overexpression provided by the HEK293 cellular system, and therefore their functional relevance is limited. All relevant functional information has been drawn from the experiments conducted in reconstituted MEFs, THP1 cells and BMDMs.

This notion is now more explicitly mentioned in a new text located in **page 8** of the revised manuscript.

POINT 4. Reviewer #1 indicated that “The claim that this process is different from the recently described LANDO, is based on differences in the kinetics of both processes. However, the recently described LANDO has been shown in a completely different system, and comparing kinetics in different cell types in response to different stimuli might be not the best way to assess whether these two processes are indeed different. More clear cut data supporting differences would be required to allow that statement (e.g. is the Rubicon-dependency different between the process described in this manuscript and LANDO?)”

We agree. This has been done.

We have used a lentiviral version of the CRISPR/Cas9 system to deplete the expression of Rubicon in MEFs expressing full-length ATG16L1 or just the Nt domain and also harbouring IL-10Rs A and B. We found that the IL-10 signalling output of these cell lines is not hampered by the absence of Rubicon. In fact, we detected a slight increase in IL-10 signalling that was comparable in both cell lines (FL and Nt). These data indicate that LANDO of IL-10Rs is unlikely to be the process whose disruption by absence of the WDD results in the defective IL-10/IL-10R trafficking and signalling features that we describe in our manuscript. The observed signalling potentiation provided by Rubicon depletion is intriguing, but its nature and mechanistic bases are currently unknown. In any event, since this effect is equally observed in FL and Nt cells, it is probably unrelated to the biology of the WDD that we address in our manuscript.

These results are included in a new **Suppl. Fig. 14a**, described in a new text included in **page 12** and further discussed in **page 17 (top)** of the revised manuscript. We have eliminated our reference to the different kinetics of both processes as a way to argue that the WDD-dependent process that we describe in our manuscript is different from LANDO.

POINT 5. The reviewer mentions that: “Whereas it is easy to understand why STAT3 (total) and pSTAT3 (cytosolic ready to be imported into the nucleus) levels should be different in the immunoblots on cytosolic extracts, it is not clear why this is also a case when using nuclear extracts, as almost all STAT3 detected in the nucleus should be pSTAT3”.

We agree.

Certainly, as the reviewer points out, if phosphorylation of STAT3 in the cytoplasm is required for its nuclear translocation, variations in the levels of nuclear unphosphorylated STAT3 should parallel those of P-STAT3 induced by IL-10, and we were surprised to see that this is actually not the case (see **Figs. 4c, 6a, 6c and Suppl. Fig. 8b**). In fact, we also detected substantial presence of nuclear STAT3 in control untreated cells that show complete absence of nuclear P-STAT3 (see control points in the same figures). Puzzled by these findings, we reviewed the available literature and we found several articles describing that unphosphorylated STAT3 is commonly present in resting cells (Meyer et al. 2002. *Exp. Cell. Res.*, 272:45. PMID: 11740864), plays active roles in the regulation of gene expression (Yang et al. 2007. *Genes Dev.*, 21:1396. PMID: 17510282) and its nuclear presence is regulated in a complex way by nuclear export (Bhattacharya et al. 2002 *J. Clin. Invest.*, 111:553. PMID:12588893). Therefore, it appears that the biology of STAT3 is not only controlled by phosphorylation but also by an additional layer of regulation that determines the basal presence of the unphosphorylated form in the nucleus.

We believe that that the high constitutive presence of nuclear unphosphorylated STAT3 that we found in our assays likely quenches how much the translocation of P-STAT3 induced by IL-10 contributes to change the total levels of nuclear STAT3. This effect likely explains why our Western-blot for total nuclear STAT3 do not parallel the changes observed for nuclear P-STAT3.

POINT 6. The reviewer indicated that *“It would also help that in those immunoblots that show differences in signaling (Figs. 4 and 5), quantifications of immunoblots and statistical analysis would be offered to better appreciate the significance of the differences in signaling events in the different situations analyzed.”*

We agree. This has been done.

We have quantified the signal intensity of the relevant bands in the Western-blot shown in **Figs. 4c, 6a, 6c and Suppl. Fig. 8b**. At the bottom of the figure panels we now provide densitometric values corrected against the relevant controls in each case (cytoplasmic P-STAT3 against total STAT3 and GAPDH, cytoplasmic IL-10 against GAPDH, and nuclear P-STAT3 against total STAT3 and Histone-3 (H3)). The numerical differences obtained solidify the results. A statistical analysis of the nuclear P-STAT3 data shown in **Fig. 6c** has also been carried out. However, due to the atypical kinetics of nuclear P-STAT3 at early times of IL-10 stimulation in the Pair #1 mouse dataset compared with the other Pairs (#2 and #3), average differences between FL and Nt are not statistically significant if the data are separately analysed for each time point. Since the dissimilar P-STAT3 intensities between FL and Nt cells mainly appear 5 and 15 min after IL-10 treatment, we pooled together the densitometric data obtained for these two time points to performed the statistical analysis. The resulting data are now statistically different, and graphs displaying these data have been included as a new **Suppl. Fig. 16c**.

In response to the request by other reviewer, we also provide quantifications of the IL-10R immunoblots shown in **Suppl. Fig. 5c and Suppl. Fig. 15c**.

REVIEWER #2

We thank the reviewer for his/her insightful comments. In the following we provide a point-by-point reply.

MAJOR CONCERN. The reviewer thinks that *“My major remark mostly concerns experimental emphasis of the paper and final message of the authors. While a considerable amount of efforts have been made in the biochemical description of the screen that lead to IL-receptors identification and in the importance of ATG16L1 WD40 domain, the final conclusion suggests that IL-10RB and IL-2R γ endo-lysosomal trafficking is a key target of IL-R/ATG16L1 interaction relevancy. This “dynamic and endosomal trafficking” hypothesis could thus benefit of additional experimental framework to propose a more detailed mechanistic explanation. While experiments using EEA1 staining are informative, this part of the message could be strengthened by dynamical analyzes of “in time/in space” IL10 endocytosis and sorting with other techniques, such as fluorescent dextran pulse-chase experiments and other endo-lysosomal markers”*.

We agree that this is a critical point. We have addressed this issue experimentally.

We started by optimizing the system to detect endosomes generated upon IL-10 exposure using a dextran 10.000 MW fixable formulation coupled to Alexa-647 (Invitrogen). Indeed, we observed formation of dextran-positive endosomes in response to IL-10, and their relative numbers in FL and Nt cells nicely paralleled those of intracellular IL-10RB-Flag-positive vesicles that we found at different times post IL-10 treatment. That is, Nt cells showed reduced numbers of dextran-positive endosomes compared to FL cells at all time points tested, including a new 30 min IL-10 treatment time that we included to increase the *“in time”* coverage of our assays. These results are now shown in a new **Suppl. Fig. 9a** of the revised manuscript, and briefly mentioned in **page 11 (top)** of the revised text.

Unfortunately, however, we were unable to co-stain the dextran-labelled vesicles for other markers, since the required permeabilization step resulted in loss of dextran signal. We tried to permeabilize the fixed cells under a variety of softer conditions, including different concentrations of NP-40 (0,5%, 0,25%, 0,1%, 0,005%) or the milder detergent saponin (0,5%, 0,1%) but none of them preserved dextran fluorescence. The reason for this technical difficulty is unclear. In order to carry out the experiments requested by the reviewer (*“in time/in space analyses” [...] “and pulse-chase experiments with other endo-lysosomal markers”*), we resorted to pulse-chase assays where we co-stained for IL-10RB-Flag and EEA1 or LAMP1 as early and late endosomal markers, respectively. We also performed similar studies under continuous IL-10 treatment for LAMP1 and Rab7, the latter as an additional late endosome marker. All analyses were extended to an additional time point (30 min) of IL-10 treatment for increased *“in time”* coverage. We found that the level of colocalization between endocytosed IL-10RB and the early endosomal marker EEA1 follows in the new pulse-chase assays the same trend observed in the continuous IL-10 treatment experiment originally shown in our initial submission (**old main Fig. 4d, now included in an independent new Fig. 5 in the revised version**). That is, we detected again a significantly reduced rate of IL-10RB-Flag/EEA1 colocalization in Nt cells early after IL-10 activation (5 and 15 min time points), and also when comparing experimental points with similar numbers of IL-10RB-Flag-positive vesicles (FL 5 min vs Nt 15 min). These new pulse-chase results are shown in a new **Suppl. Fig. 10**, and are briefly described in **page 11** of the revised manuscript. On the other hand, FL and Nt cells showed similar colocalization rates between IL-10RB and LAMP1 (both in continuous IL-10 treatment and pulse-chase studies) or Rab7 (only continuous treatment) compared to FL cells, both in the 15 min time point and also when comparing FL 5 min with Nt 15 min data. We only detected decreased colocalization rates of these late endosome markers with IL-10RB in Nt cells in the 5 min post-IL-10 treatment point, where there are almost no IL-10RB-positive endosomes in Nt cells and therefore this parameter is probably irrelevant. These new data for LAMP1 and Rab7 are shown in **Suppl. Figs. 11-13** and described in **page 11** of the revised text.

In our view, these results show that absence of the WDD causes a defect in the trafficking of IL-10/IL-10R complexes to early endosomes, whereas their targeting to late endosomal/lysosomal compartments remains for the most part unaffected. Consequently, these results appear to exclude that the IL-10/IL-10R endosomes are degraded at a faster pace in Nt cells, because IL-10R trafficking to the lysosomes (LAMP1-positive signal) is not favoured in these cells. Therefore, enhanced IL-10R endosome degradation does not seem to be the main alteration that explains why the number of IL-10R-positive endosomes is diminished in cells lacking the WDD. Interestingly, these data also indicate that IL-10R endosomes do not access the lysosomal compartment more efficiently in FL cells either, where they are however more prominently labelled with LC3. Consequently, these results also appear to argue against a possible degradative role of LC3 in this context (see below for additional considerations regarding this point). More generally, the data indicate that endocytosed IL-10Rs reach the same intracellular compartments in both cellular strains, thus pointing to the idea that absence of the WDD does not cause trafficking of the engaged IL10Rs to substantially different subcellular compartments. This is, of course, considering the range of markers that we have used, which does not allow us to formally exclude the possibility of a differential trafficking to other unidentified compartments. More systematic studies with a wider range of subcellular markers would be required to answer this question, albeit with a substantial risk since the nature of such possible compartments is currently unknown.

Although not specifically requested by the reviewer, we have carried out an additional experiment that we think provides relevant information in this regard. In order to explore in a different way whether the decreased number of IL-10R-Flag-positive vesicles in Nt cells comes from reduced endocytosis of activated IL-10Rs or increased degradation of IL-10/IL-10R-positive endosomes, we conducted a sort of “IL-10R endocytic flux” assay where we stimulated the system with IL-10 (in a pulse-chase modality) in the absence or presence of bafilomycin A1 to inhibit lysosomal activity, and then evaluated the number of intracellular IL-10RB-Flag-positive vesicles per cell in the different conditions. We reasoned that inhibition of lysosomal degradation by bafilomycin would eliminate any contribution of a possibly dissimilar degradation rate of IL-10R endosomes in FL and Nt cells, thus revealing the real contribution of endosome generation to the whole phenotype. In these experiments we found that bafilomycin does not normalize the different number of IL-10RB-Flag endosomes observed in FL and Nt cells, indicating that such difference likely arises for the most part from a reduced rate of endocytosis and/or early endosome formation in Nt cells. This result is consistent with the observation that IL-10Rs do not show enhanced colocalization with LAMP1 in Nt cells. Importantly, bafilomycin increased the number of Flag-positive vesicles in all conditions, indicating that some degree of IL-10R endosome degradation is taking place. The new bafilomycin experiments are shown in **Supp. Fig. 9b** and described in **page 11 (top)** of the revised text.

Therefore, in response to the specific request of the reviewer that we “*propose a more detailed mechanistic explanation*” to provide a more clear “*final message*”, we believe that the main role of the WDD in IL-10R biology is to favour endocytosis and early trafficking of IL10/IL-10R endosomes, without having a major impact on their lysosomal degradation or in the nature of the endocytic route they follow on their way to the lysosome. Still, as mentioned above, we cannot exclude that the WDD could promote their transit to unidentified subcellular compartments that might enable optimal signalling. Such increased endocytosis and early trafficking (in the absence of a dissimilar degradation rate) causes enhanced accumulation of endosomes harbouring activated IL-10Rs in FL cells, which, considering the view that endosomes constitute important signalling platforms, likely results in improved IL-10 signalling output. Reversing the argument, absence of the WDD results in a substantial

reduction in the total number of endosomes harbouring activated IL-10/IL-10R complexes mainly by reducing their endocytic input into the pathway, thus causing the impaired IL-10 signalling output that we observe in Nt cells.

How ATG16L1 may promote endocytosis of IL-10Rs is currently unclear. A previously described interaction between the N-terminal region of ATG16L1 and the clathrin heavy chain (Ravikumar et al, 2010) could promote endocytosis upon recruitment of ATG16L1 to the activated IL-10Rs, but a possible role of LC3 lipidation in this scenario would not be obvious. In this context ATG16L1 might function mainly as a scaffold protein, independently of its LC3 lipidation ability. How ATG16L1 may favour efficient early trafficking of IL-10R endosomes is also unclear but, interestingly, in addition to the degradative role of LC3 in a variety of pathways (ranging from canonical autophagy (Bento et al, 2016) to LAP (Sanjuan et al, 2007)), LC3 is also known to favour membrane fusion events (Weidberg et al, 2011), including, for example, those involved in the secretion of lysosomal contents through the osteoclast ruffled border (DeSelm et al, 2011). A role of LC3 in membrane fusion could facilitate the transition of LC3-labelled IL-10R endosomes to EEA1-positive vesicles, a scenario that would point again to a non-degradative function of LC3 in the process that we describe in our study. Such atypical non-degradative function of LC3 may be implicit in our model, since the canonical degradative role of LC3 would be expected to increase fusion of LC3-labelled IL-10R endosomes with the lysosomal compartment to cause earlier signal cessation, instead of an enhanced output. Exploration of these issues, and whether or not LC3 is truly involved in the mechanism that we describe in this manuscript, are certainly very interesting themes that we will address in subsequent studies. More generally, our results underscore the notion that timely endocytosis and proper endocytic trafficking are important factors that maximize the overall signalling output of cytokine receptors. All these considerations are mentioned in **pages 15-16** of a substantially expanded Discussion section now included in the revised manuscript.

For a more linear argumentation, in the revised manuscript we have segregated the figures showing just vesicle number data (**now in a new Fig. 4d**) from those showing colocalization assays with EEA1 (**now included in an independent new Fig. 5**), which were originally shown together in the context of the colocalization studies with EEA1 (**old Fig. 4d**). This reorganization resulted in one additional main figure showing the EEA1 trafficking data (**new Fig. 5**). In our view, this slightly modified structure allows easier accommodation of the new dextran (**Suppl. Fig. 9a**), bafilomycin (**Suppl. Fig. 9b**) and marker colocalization (**Suppl. Figs. 10-13**) data in the overarching logic of the manuscript.

MINOR POINT 1. Reviewer #2 thinks that *“While I found that the paper is very well written, I think that it could benefit from a more detailed discussion section, considering the important amount of data to be discussed.”*

We agree. This has been addressed.

The manuscript was originally written for a shorter format. Following the formatting policy of *Nat. Commun.*, we have divided the original manuscript into formal Introduction, Results and Discussion sections and included sub-headings in the Results part. In addition, as suggested by the reviewer, the Discussion section was substantially expanded to accommodate a number of considerations regarding the mechanistic consequences of our data, especially in light of the new results provided in the revised manuscript, and how the process we describe is different from other unconventional autophagic activities. We also discuss the implications that the new WDD-binding motif and the identified collection of type-I transmembrane proteins may have in the future expansion of the field, in addition to the remarks that were already discussed in the original version of the manuscript.

The new, richer Discussion section is included in **pages 15-18** of the revised manuscript.

MINOR POINT 2. The reviewer indicated that “*In line with this – and this is only a suggestion – the title could mention “endosomal cytokine signalling” rather than simply “cytokine signalling”.*”

We respectfully disagree.

We modestly believe that the title is already somewhat long (16 words, while the allowed title length is just 15 words) and complex, so that adding “endosomal” would make it a little bit more convoluted and hard to understand, although we agree with the reviewer that the modified title would convey the point of the article more accurately. The idea that the WDD regulates IL-10R signalling by controlling its endosomal trafficking is explicitly mentioned in the Discussion section (**page 15**).

MINOR POINT 3. The reviewer indicated that “*The section about “autophagic activity” of IL-2R, IL-10R and IL-33R is not very clear*”.

We agree. This has been corrected.

We have expanded this section to clarify the rationale and methodological details of the experimental approaches that we used to characterize the autophagic activity of the identified receptors. In particular, we mention that these approaches were implemented based on our previous experience on TMEM59. Since straight overexpression of TMEM59 suffices to induce unconventional LC3 lipidation, we used this simple property to quickly identify receptors able to engage ATG16L1 with functional consequences. We also explain with more detail why cells expressing separated Nt and Ct domains of ATG16L1 are useful to establish if the autophagic activity of the receptors is truly unconventional in nature (that is, WDD dependent). Finally, we describe that an unconventional autophagic activity triggered through direct ATG16L1 engagement in the vicinity of a membranous source for LC3 lipidation is expected to cause LC3 labelling of the same vesicles where the inducer is located, just like it happens in the case of TMEM59.

The original text has been modified to accommodate these considerations and is now located in **page 7** of the revised manuscript version. We hope this section is now more clearly presented.

MINOR POINT 4. The reviewer indicated that “*Could the authors precise with more details why they focused on IL-10RB associated signalling in this context?*”.

We agree. This has been done.

We now provide a more detailed explanation of why we chose to focus on IL-10RB for further studies to explore a possible role of the WDD in IL-10 signalling. We did so because there is an intriguing functional link between the main physiological activity of IL-10 (the control of inflammatory responses) and a few reports indicating that ATG16L1 and the WDD have anti-inflammatory roles in different experimental settings (see, for example, Saitoh et al. Nature, 456:264 (2008) or Sorbara et al. Immunity, 39:858 (2013), both articles included in our manuscript). Given this connection, we hypothesized that perhaps the WDD could allow ATG16L1 to support IL-10 signalling, a possibility that might explain some of the anti-inflammatory roles of ATG16L1.

New text briefly describing these considerations has been included in **pages 8 (bottom) and 9 (top)** of the revised manuscript. This idea is mentioned again in the Discussion section (**page 17, bottom**).

MINOR POINT 5. The reviewer mentions that “It is not clear to me why the LC3 WB shown in Sup Fig5b only displays one band”.

We agree.

In our experience, endogenous LC3-I is often right at the limit of detection, and therefore usually produces very faint bands irrespective of how strong the LC3-II signal may be. Faint or non-existing endogenous LC3-I bands can also be observed in **Suppl. Fig. 5a** and in **Suppl. Figs. 15a and 15b** (the latter showing similar studies for THP1 cells). This is in contrast to assays involving transfected LC3, where LC3-I is always readily detectable (**see, for example, Figs. 2b or 3c**). Why endogenous LC3-I generates such low signals is unclear, but in our hands, it is not unusual that, being at the verge of detectability, some assays provide sufficient LC3-I signal to produce a very light band (see the E64d/pepstatin panel in **Suppl. Fig. 5b**), whereas in other cases the band cannot be detected at all, as it occurs in the bafilomycin panel the reviewer refers to (**Suppl. Fig. 5b**), or in the bafilomycin-treated points shown in **Suppl. Fig. 15b** (THP1 cells).

MINOR POINT 6. The reviewer indicates that “The increased levels of IL-R in ATG16L1 FL or Nt cells shown by western blot in Sup Figure 5c could benefit of quantification(s)”

We agree. This has been done.

We now include quantifications of the IL-10R immunoblots shown in **Suppl. Fig. 5c** and **Suppl. Fig. 15c**. Densitometric data were corrected against the signal provided by the loading control (GAPDH). These numerical data confirm comparable expression levels for both IL-10RA and B between FL and Nt cells. However, the expression rate of both receptors is increased in ATG16L1^{-/-} cells, a phenomenon that we currently cannot explain.

In response to a request by other reviewer, we also include quantifications of the Western-blots shown in **Figs. 4c, 6a, 6c and Suppl. Fig. 8b**.

MINOR POINT 7. The reviewer mentions that “Surface Exposure analyzes of IL-10R where analyzed by flow cytometry. Is that possible to show similar experiments in immunofluorescence?”

We agree. This has been done.

We have performed immunofluorescence studies to evaluate surface IL-10R expression by confocal microscopy. The same anti-IL-10R antibodies used for flow cytometry were used to stain unpermeabilized FL and Nt cells expressing both IL-10Rs. Cells were then fixed and processed for immunofluorescence and confocal microscopy. We provide a new panel as **Suppl. Fig. 5e** showing representative confocal pictures for both IL-10Rs in both FL and Nt cells. We did not detect any difference in the staining pattern between the 4 different conditions (FL and Nt cells for both IL-10RA or IL-10RB).

MINOR POINT 8. The reviewer indicates that “As LC3 is not only present on (transient) autophagosome, but also on pre-autophagosomal structures (omegasomes,

phagophores) as well as autolysosomes, the notion of “amphisome” which is proposed by authors (Sup Figure 7) is not very convincing in this context. Amphisomes existence is still under debate and thus needs, at a minimal, a full characterization using biochemical and morphological precise techniques”.

We agree.

We have now eliminated all references to amphisomes throughout the manuscript.

REVIEWERS' COMMENTS

Reviewer #1 (Remarks to the Author):

In this revised version, the authors have addressed all concerns raised in the original submission. New data provides a deeper understanding of how the interaction of ATG16L and the IL-Rs modifies their incorporation into early endosomes w/o altering their degradation rates, and more clearly identify this process as different from LANDO. Furthermore, new experiments also offer conclusive data that characterizes the effects that the interaction of the WD40 domain of ATG16L and the IL10RB has on signaling activated downstream on that IL receptor. The manuscript offers now compelling evidence that identifies a new role for an autophagy protein, ATG16L, that is not dependent on its function on canonical autophagy and may likely be independent of its ability to induce LC3 lipidation. The regulation of the trafficking of the IL10R by this mechanism likely extends to other receptors, including the IL2R. Indeed, the method developed by the authors to identify new interacting partners of the WD40 domain of ATG16L will likely lead to the characterization of other physiological processes that are also regulated by this mechanism.

Reviewer #2 (Remarks to the Author):

Authors addressed and/or replied to my questions and remarks and the revised version of the manuscript is more robust in the present form.

In the following we address the specific comments of the reviewers.

REVIEWER #1

POINT 1. Reviewer #1 mentions that “In this revised version, the authors have addressed all concerns raised in the original submission. New data provides a deeper understanding of how the interaction of ATG16L and the IL-Rs modifies their incorporation into early endosomes w/o altering their degradation rates, and more clearly identify this process as different from LANDO. Furthermore, new experiments also offer conclusive data that characterizes the effects that the interaction of the WD40 domain of ATG16L and the IL10RB has on signaling activated downstream on that IL receptor. The manuscript offers now compelling evidence that identifies a new role for an autophagy protein, ATG16L, that is not dependent on its function on canonical autophagy and may likely be independent of its ability to induce LC3 lipidation. The regulation of the trafficking of the IL10R by this mechanism likely extends to other receptors, including the IL2R. Indeed, the method developed by the authors to identify new interacting partners of the WD40 domain of ATG16L will likely lead to the characterization of other physiological processes that are also regulated by this mechanism”.

We appreciate the kind comments of the reviewer and his/her suggestions to improve our manuscript.

REVIEWER #2

POINT 1. The reviewer thinks that “Authors addressed and/or replied to my questions and remarks and the revised version of the manuscript is more robust in the present form.

We thank the reviewer for his/her insightful comments and suggestions to improve our manuscript.